# Evaluation of ALARO-0 and REMO Regional Climate Models over Iran Focusing on Building Material Degradation Criteria

Hamed Hedayatnia [1,*] , Sara Top [2] , Steven Caluwaerts [3,4], Lola Kotova [5], Marijke Steeman [1] and Nathan Van Den Bossche [1]

1   Building Physics Group, Faculty of Engineering and Architecture, Ghent University, 9000 Ghent, Belgium;
    Marijke.Steeman@UGent.be (M.S.); Nathan.VanDenBossche@UGent.be (N.V.D.B.)
2   Department of Geography, Faculty of Sciences, Ghent University, 9000 Ghent, Belgium; sara.top@ugent.be
3   Atmospheric Physics Group, Faculty of Sciences, Ghent University, 9000 Ghent, Belgium;
    steven.caluwaerts@ugent.be
4   Department Meteorological and Climatological Research, Royal Meteorological Institute of Belgium,
    1180 Brussels, Belgium
5   Climate Climate Service Center Germany (GERICS), Helmholtz-Zentrum Hereon, 20095 Hamburg, Germany;
    lola.kotova@hereon.de
*   Correspondence: hamed.hedayatnia@ugent.be; Tel.: +32-496026475

**Abstract:** Understanding how climate change affects material degradation is the first step in heritage conservation. To study such impact, high-resolution climate information is required. However, so far, no regional climate simulations have been evaluated considering building damage criteria over the region of Iran. This paper has a twofold objective: to conduct an overview of climate model performance over Iran by evaluating the output of two regional climate models, ALARO-0 and REMO2015, and to find an optimal approach for model evaluation fitted to studies on building physics. Data of the evaluation run for both models were compared with data of weather stations located in six different climate zones in Iran to assess their performance over the region and gain insight about model uncertainties. Given that the research scope covers the evaluation of climate models to use in studies on building physics, in addition to climate parameters, five degradation risks are analysed. The performance of the two models varies over the studied locations. In general, both models fall within the spread of observations except for wind parameters. Accordingly, indices related to temperature and precipitation are well predicted, in contrast to indices related to wind. The analysis shows that considering the observed biases, selecting an ensemble of representative models based on the evaluation results of climate variables important for hygrothermal simulations would be recommended.

**Keywords:** climate change; heritage; building damage criteria; regional climate model; ALARO-0; REMO2015; model evaluation; Iran; hygrothermal simulations; model evaluation

## 1. Introduction

The Iranian plateau hosts one of the oldest civilisations in the world. The country's rich cultural heritage is reflected by its 22 UNESCO world heritage sites [1]. The climate of Iran is diverse, ranging from arid and subtropical conditions over most of the country to mild conditions along the Caspian coast in the north and cold conditions in the mountain ranges, such as the Zagros and Alborz Mountains [2].

Today, the study of historical observations shows a clear and concerning change in global climate. Heritage sites can be vulnerable to changes in weather patterns [3]. The Fifth Assessment Report of the Intergovernmental Panel on Climate Change [4] suggests even more significant changes in regional climate conditions during the next century, such as drier and hotter summers (JJA) over the Middle East. Since it is unknown what the impact of this climate change will be on heritage sites, this should be assessed. Thus, so

far, no relevant research has been performed over the Iranian plateau focusing on the current and future climate change effects on building material degradation. This study aims to evaluate two regional climate models over the Iranian plateau, considering building material degradation criteria.

Temperature is a fundamental parameter for heritage sites, since it has a significant impact on material durability. For example, higher temperatures cause reduced material stiffness and strength. Knowing the average ambient temperature is vital in calculating various building material degradation indices such as freeze–thaw cycles (FTCs), salt crystallisation and the moisture index (MI).

FTCs are critical degradation criteria for building materials, particularly for historical buildings due to material ageing. Hence, the minimum temperature is relevant during both historical analysis and model validation to know to what extent models are able to reproduce low temperatures. Additionally, high temperatures are among the most critical factors that can change the thermal properties of building materials and cause thermal stress in building components. Therefore, the average ambient temperature and minimum and maximum temperatures of the models were evaluated in this study.

The presence, accumulation and periodic variations of moisture in the mass and on the surface of building components affect the building's energy efficiency and cause various types of deterioration and degradation mechanisms. Specifically, the accumulation of moisture in building materials (resulting from water vapour condensation, rainwater penetration, groundwater uptake, etc.) increases the thermal conductivity and decreases the insulation capacity [5–13]. Additionally, changes in moisture content can cause swelling and shrinkage of building components, as well as solution and precipitation of salts inside the building materials, which can cause strains and cracks in the components. Salt weathering is among the critical degradation criteria in heritage sites, and it is driven by phase changes due to variations in relative humidity that depend on the local climate.

Further, assessing the MI is essential to study the process of building material deterioration. Mould and decay in building materials are critical with regard to building durability and are caused when the humidity level exceeds the tolerance of the structure [14]. Hence, an assessment of model reliability for this factor is necessary. In addition, aging of the materials is affected by vapour pressure, relative humidity and precipitation. These climatic parameters play essential roles in the erosion of the building envelope.

Furthermore, there is a significant impact of wind-driven rain, which induces damages to and erosion of the building envelope. Therefore, simulated wind must be evaluated before it is used as input in building erosion assessment. In order to study the future climate and its impacts, data of climate simulations are needed. General circulation models (GCMs) are used to study climate conditions on a global scale. However, the spatial resolution of such global simulations, generally not less than 50 km [15], is too coarse to identify climate variability on a regional scale, e.g., resulting from orography. Through regional climate models (RCMs), GCM data can be downscaled to higher resolution across a specific region. Using climate models is the first step in assessing what the future will bring; however, these models need to be validated.

Data of the REMO and ALARO-0 regional climate models have been generated by the AFTER project (https://www.projectafter.net) and accessed on 25 May 2020. The project, funded by the ERA.Net RUS Plus Initiative, ID 166, aims to foster research cooperation between Russia, the European Union and Turkey. Both regional models are being developed by the partner institutions of this project. In the AFTER project's evaluation of ALARO-0 and REMO at 0.22° horizontal resolution, the models showed good performance over western Central Asia, including Iran [16].

In studies on building physics, climate projections are often used without any specific validation, hence it is unclear how reliable the associated hygrothermal output is. In addition, typically only a single GCM-RCM combination is adopted. In this study, two RCMs are used and compared with observations to assess the model-specific bias effects that may arise. Given that both models have previously been validated for that specific

region from a climatological point of view, it can be considered as the best-case starting point for studies on building physics when no specific validation has been performed that considers the material degradation risks.

Since it is crucial to know the future climate and its uncertainties at the locations of heritage sites, evaluations were conducted at point locations instead of doing an evaluation study based on comparisons with gridded observational datasets.

In this paper, ALARO-0 and REMO, which were well evaluated over the region, are assessed for selected locations. Different climatic parameters and indices are analysed based on a dataset of 38 years (1980–2017) for six locations across the Iranian plateau.

Three-hourly data are essential for studying daily variability, which plays a critical role in, e.g., FTCs and thermal stress, whereas monthly and seasonal intervals are used for slower processes such as interstitial condensation. Annual time series are critical to identify temporal trends and their significance [17]. Finally, a unique methodology was used to evaluate the output of the RCMs. A comparison between the observed minimum, mean and maximum temperature and the modelled data across the studied locations is presented for the studied period, together with an analysis of FTCs (Section 3.1). Section 3.2 presents the results for the relative humidity and salt crystallisation. Section 3.3 presents an evaluation of the modelled precipitation parameters and MI. Section 3.4 assesses the accuracy of the model for wind velocity and direction across the studied locations. The last section evaluates wind-driven rain, followed by the conclusion.

## 2. Materials and Methods

### 2.1. Study Area

Given that Iran has an extended range of climate regimes, the analysis was performed over 6 meteorological stations in different climatic zones, as shown in Figure 1 and Table 1. It is clear from the map that the study locations are characterised by different climates and environments. The capital city, Tehran, experiences a cold wet climate in winter (DJF), a cold semi-arid climate with a mild climate during spring (MAM) and autumn (SON), and a hot-dry climate in summer (JJA).

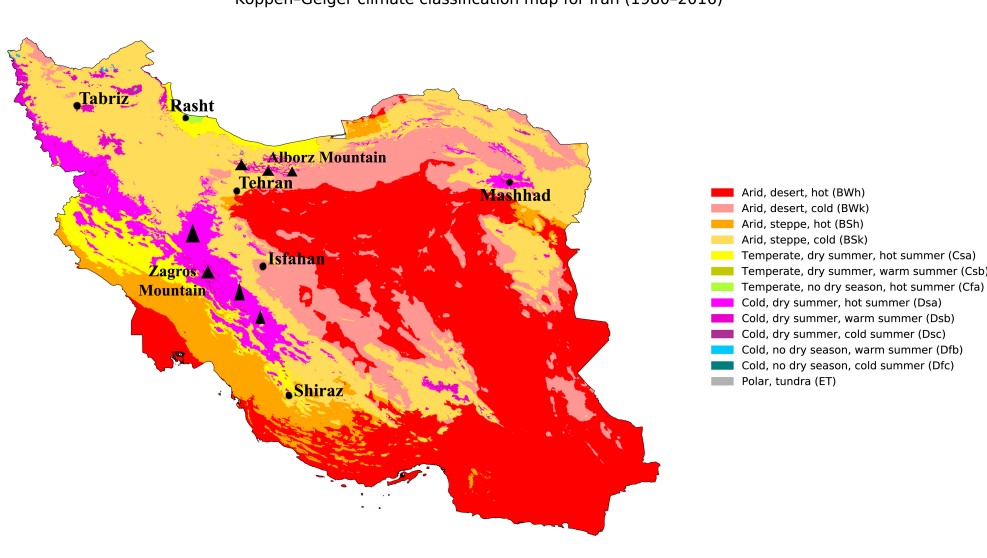

Köppen–Geiger climate classification map for Iran (1980–2016)

Source: Beck et al.: Present and future Köppen-Geiger climate classification maps at 1-km resolution, Scientific Data 5:180214, doi:10.1038/sdata.2018.214 (2018)

**Figure 1.** Köppen-Geiger climate classification map of Iran, with studied locations and mountainous areas.

**Table 1.** Overview of 6 point locations and their climate (1980–2017).

| Location | Tehran | Mashhad | Shiraz | Tabriz | Rasht | Isfahan |
|---|---|---|---|---|---|---|
| Coordinates | 35.41° N/51.19° E | 36.2° N/52.6° E | 29.6° N/52.6° E | 38.1° N/46.2° E | 37.3° N/49.6° E | 32.5° N/51.7° E |
| Orographic features | Alborz Mountains to the north and central desert to the south | Valley of Kashafrud River, between two mountain ranges of Binalood and Hezar Masjed | Shiraz plain surrounded by mountain ranges with an average height of 2000 m | Located between Eynali and Sahand mountains in a fertile area | City on Caspian Sea coast | Situated at foothills of Zagros mountain range |
| Altitude (m) | 1191 | 999.2 | 1488 | 1361 | −8.6 | 1550.4 |
| ALARO-0 altitude (m) | 1485.45 | 1343 | 1749 | 1682 | 16.7 | 1705 |
| REMO altitude (m) | 1109.7 | 1087 | 1723 | 1567 | −1 | 1683 |
| Climate | Bsk/Csa/Dsa | Bsk | BSh/Bsk | Dsa | Csa | BSk |
| Mean Tmin (°C) | 10.5 | 7.3 | 9.2 | 7.2 | 11.1 | 9.4 |
| Mean Tmax (°C) | 20.4 | 21.2 | 25.6 | 18.2 | 20.5 | 23.1 |
| Annual precipitation (mm/year) | 429.2 | 251.5 | 305.6 | 318.8 | 1255.5 | 114.3 |

Tehran is located on the Alborz hillside, and Tochal Mountain, at 3974 m height, overlooks the north of the city. The measurement station is located in the north of the city. Mashhad is characterised by a steppe climate with a hot summer (JJA) and a cold wet climate in winter (DJF). Mashhad is the second largest city in Iran, characterised by a dense metropolitan area and fast urban sprawl, particularly during the 1980s. Shiraz is located in the south of Iran and experiences a hot semi-arid climate. Tabriz has a humid continental climate with regular seasons bordering a cold semi-arid climate. Rasht is situated between a humid subtropical climate and a Mediterranean climate. Isfahan is located in the plain of the Zayanderud River in the centre of the Iran plateau and experiences a cold desert climate.

## 2.2. Model Description and Experimental Design

The RCM data in this analysis originated from ALARO-0 and REMO models running at a spatial resolution of 25 km across central Asia at an hourly time frequency. The ALARO-0 model version has been described in the referenced articles [17–19]. The most recent version, REMO2015, has been described in the referred literature [20]. The evaluation runs of both RCMs were used in the current study. The RCM experiments were forced by lateral boundary conditions coming from the ERA-interim reanalysis.

The evaluation analysis in the current study was performed by comparing observational data of the six locations with the modelled data over thirty-eight years (1980–2017) for the meteorological variables temperature, precipitation, relative humidity, wind velocity and wind direction, as well as for derived building material degradation indices: freeze–thaw index, salt crystallisation and MI. The nearest-neighbour method was applied to obtain the model data of the closest grid point to the six selected locations (Table 2). For both ALARO-0 and REMO, hourly values of 2 m temperature, 10 m wind speed and precipitation, as well as 3-h values for relative humidity and wind direction, are available. Daily, monthly and yearly averages for temperature, relative humidity and wind velocity were calculated from these hourly values. The average daily values of temperature and relative humidity were used to calculate daily mean vapour pressure. The hourly values for temperature were used to compute daily minimum and maximum temperature. A height correction based on differences in topography between the model and the observation points and assuming a uniform temperature lapse rate of $0.0064 \text{ k} \cdot \text{m}^{-1}$ was applied for the model temperature. For precipitation, the accumulated values for daily, monthly and yearly time intervals were considered. For the observations, meteorological data provided by the National Weather Service recorded for the same parameters taken from the three-hourly

instantaneous fields were used to compute the minimum and maximum daily, monthly and yearly temperature and mean values the other parameters. For precipitation, accumulated values were considered. Observed and modelled parameters with their temporal resolution for the six locations are summarised in Table 2.

**Table 2.** Overview of meteorological parameters and time resolution for observations and model data.

| Reference Parameters | Air Temperature | Precipitation | Relative Humidity | Wind Velocity | Wind Direction |
|---|---|---|---|---|---|
| Temporal resolution observations | 3-h<br>2 m | 6-h | 3-h | 3-h<br>at 10 m | 3-h<br>at 10 m |
| Temporal resolution model data | 1-h<br>2 m | 1-h | 3-h | 1-h<br>at 10 m | 6-h<br>at 10 m |

To explain the significant bias over Tehran, the gridded Climatic Research Unit (CRU) TS dataset (version 4.02), which contains 10 climate-related variables for the period 1901–2018 at a grid resolution of 0.50°, was used as a reference.

### 2.3. Method of Analysis

### 2.3.1. Statistical Analysis

The evaluation was performed by processing and comparing the models' outputs with the observed data. The assessment results were visualised using probability density function (PDF) plots (kernel distribution) for annual cycles with daily frequency, annual trends and Taylor diagrams. Taylor diagrams provide a way to graphically summarise how closely a pattern or a set of patterns matches the observations. In the Taylor diagrams, the model reliability for the main parameters is quantified in terms of temporal correlation, centred RMSD and the ratio of temporal variability between the model and the observational dataset [21]. Standard deviation is used to quantify the variability, and normalisation is obtained by taking the ratio. In the normalised Taylor diagram, the perfect model lies at a correlation of 1 and RMSD of 0, which means that models predicted the observational data. A normalised standard deviation closer to 1 during all seasons means that the model can better capture variations in seasonality.

Next, by analysing the yearly time series, the models' trends were studied and compared with the observations. The trend line was plotted for visual analysis. The large variability in climate leads to significant uncertainties in the trend line and large bounds. The bounds are not included for clarity.

### 2.3.2. Scoring Methodology

The calculated RMSD values were picked as comparison criteria for the main climatic parameters. Given that there is no specific set of criteria for the RCM evaluation considering material-specific indices and hygrothermal studies, the average values during the analysed period were selected to assess the models' performance with derived parameters. The modelled average values were normalised by dividing by the observed values in order to rank the models by derived parameters and indicate the model performance: a value close to 1 indicates good model performance. Moreover, the trend slope for the annual number of events was computed to evaluate the models' annual trend consistency compared to the observations.

### 2.3.3. Calculation of Indices

### Freeze–Thaw Cycles (FTCs)

The first index is the number of FTCs [22]. Based on daily mean temperature, a cycle is counted each time the temperature drops below 0 °C, given that the previous day was a non-freezing day. The annual number of FTCs for both models was computed and compared with the observations, and the annual trend of FTCs during the studied period was examined.

### Salt Crystallisation

Second, salt weathering was studied, which is one of the critical degradation criteria in historical heritage sites, driven by phase changes in relative humidity. This damage arises during salt crystallisation–dissolution cycles. Therefore, the monthly and yearly numbers of halite and thenardite–mirabilite transitions were analysed using criteria proposed in the referenced articles [23–27]. The number of phase transitions was used to estimate potential salt damage [25]. In the case of sodium chloride salt (halite), this is assessed by counting the number of times the average daily relative humidity crossed the critical deliquescence point of 75.3% for consecutive days. Only transitions that occur when the humidity decreases and crystallisation occurs are counted, and this is equivalent to the number of cycles [27].

In the case of hydrated salts, thenardite–mirabilite transitions for sodium sulphate ($Na_2SO_4 \rightarrow Na_2SO_4 \cdot 10H_2O$) are only accumulated on consecutive days when thenardite can convert to mirabilite (through thenardite dissolution followed by mirabilite crystallisation) and exert crystallisation pressure higher than 10 MPa [25]. The phase transition in salts within the building materials occurs during moisture fluctuations. The value of 10 MPa was chosen because it usually exceeds the tensile strength of porous stone. This threshold is taken from an analysis of data from the Building Research Establishment for porosity, bending strength and weight loss during salt crystallisation of British limestones. This analysis shows that damage by sodium sulphate mainly occurs in stones with 10% porosity and bending strength of 10 MPa [26]. If the temperature is lower than 22.5 °C, the Correns equation suggests a crystallisation pressure of mirabilite higher than 10 MPa [26]. The humidity of these transitions is determined from water activities, as reflected in [25], and yields a mirabilite–thenardite phase boundary of:

$$RH_{eq} = aH_2O = \sqrt[10]{K_{mir}/K_{the}} \tag{1}$$

where $aH_2O$ is the water activity and $K_{mir}$ and $K_{the}$ are the solubility products of mirabilite and thenardite [26]. This yields a 10 MPa phase boundary at:

$$RH_{eq} = 59.11 + 0.87549 \, T \text{ when } T < 22.5 \, °C \tag{2}$$

Since the stone's buffering effect in the humidity transfer was taken into account using daily mean relative humidity, the damage was slightly greater in one-day cycles [27]; thus daily values were considered.

### Moisture Index (MI)

The MI, as mentioned, is essential to study the level of wetting or drying of the construction and can be used to analyse the effects of climate change on the moisture content of building materials. Besides the MI calculated for each year for the models and observations [28], the moisture reference years (MRYs) were compared for different locations (see Supplementary Materials). The MRY selection methodology used in our research was proposed in the referenced literature [27] and uses a climate-based approach independent of the wall construction. The selected MRYs are represented in Supplementary Materials.

By using the MI, it is possible to classify individual years as wet or dry. The hypothesis is that the higher the MI, the greater the potential for moisture loading. Wet and dry years are defined as those years that deviate more than one standard deviation from the mean MI value of the sample set for a city [27].

The moisture index is defined as the ratio of the wetting index (WI) to the drying index (DI):

$$MI = WI/DI \tag{3}$$

The total annual average precipitation, average yearly rainfall multiplied by average wind velocity (aWDR) and accumulated yearly wind-driven rain can be considered to calculate the wetting index (WI). Here, the annual average precipitation is considered due to the models' significant bias for wind parameters (correlation lower than 0.50; see Section 3).

The DI is calculated for climate data using Equation (4):

$$DI = (1/n) \sum_{i=1}^{n} \sum_{h=1}^{k} \Delta W \tag{4}$$

where DI is the drying index in kilograms of water per kilogram of air, n is the number of considered years, and k is the number of hours in a particular year. Here, the drying index for each year is computed, so n is equal to 1. $\Delta W$ is the difference between the humidity ratio at saturation and in ambient conditions.

$\Delta W$ at time t can be computed using Equation (5):

$$\Delta W(t) = Wsat(t) - Wout(t) \tag{5}$$

where $\Delta W(t)$ is the difference between the humidity ratio at saturation, $W_{sat}$, and the humidity ratio at ambient conditions, $W_{out}$, at time t.

The humidity ratio (W) can be calculated using Equation (6):

$$W = 0.622 \times (v_p/p - v_p) \tag{6}$$

where W is the humidity ratio in kilograms of water per kilogram of air, $v_p$ is the vapour pressure in kilopascals (kPa), and p is the total mixture pressure in kilopascals (kPa).

Wind-Driven Rain (WDR)

Wind-driven rain is one of the critical moisture sources for the building envelope, and its quantity is an essential parameter for heat–air–moisture (HAM) analysis. Before using this climate variable in hygrothermal simulations, it is necessary to know the model's reliability for this parameter. Similar to the MI, there are different methods to calculate this parameter. Here the ISO semi-empirical method [28] is used, which is calculated based on wind velocity, wind direction and horizontal rainfall. WDR can be calculated for a specific orientation using the following Equation:

$$WDR = 2/9 \cdot V(10) \cdot \cos(\theta) \cdot (r_h)^{0.88} \tag{7}$$

where WDR is the free field wind-driven rain, V(10) is the wind speed (m/s) at 10 m height, $\Theta$ is the angle between a line normal to the wall of interest and the wind direction and $r_h$ is the rainfall on a horizontal surface (mm/h).

## 3. Results

### 3.1. Temperature

3.1.1. Mean, Minimum, and Maximum Temperature

Figure 2 shows the annual trends for minimum, mean and maximum temperature over the studied stations. The observations and models have a warming trend, but the trend is smoother for REMO and ALARO-0. The most substantial difference between the annual trends can be observed in Mashhad. Given the intense urbanisation of this city, the urban heat island (UHI) phenomenon can induce an additional warming trend at this location [29]. The fact that the most significant slope difference between model and observation can be seen for the minimum temperature supports this hypothesis, as the UHI is mainly a nocturnal effect. As the land use is kept constant during the climate runs, the increasing urbanisation is not reflected in the modelled climate, which explains the less steep trend of the model in mentioned cities [30]. Figure 3 shows the PDFs of the observed daily temperature for each model output compared to the observations for the six locations. A good overall representation of the observed temperature by both models can be seen over the studied stations, except for Tehran, where the cold bias caused a shift in the distribution of ALARO-0. ALARO-0 underestimates extreme cold and mid-range temperatures and overestimates higher temperatures, except for Shiraz, where

there is an underestimation of warm temperatures (Figure 3). REMO simulates warmer low temperatures and overestimates the presence of mid-range temperatures for Tabriz, Mashad and Rasht. A bias for warm temperature is simulated by REMO across all studied locations, except for Tehran, with a very cold bias (Figures 2 and 3; see Supplementary Materials). The reason can be seen in Figure 4: ALARO-0 data extracted for Tehran do not represent the observed climate well, since, for this location, the nearest point of the ALARO-0 grid is situated in the mountainous area to the northwest of the city, with a cold semi-arid climate (Figure 4), and shows a very cold local bias compared with REMO output and the CRU dataset (Figure 4). The obtained bias at this location results from the highly heterogeneous terrain in the gridcell with a resolution of 25 km and clearly shows the limitation of using this resolution. Using higher-resolution climate data could overcome this issue.

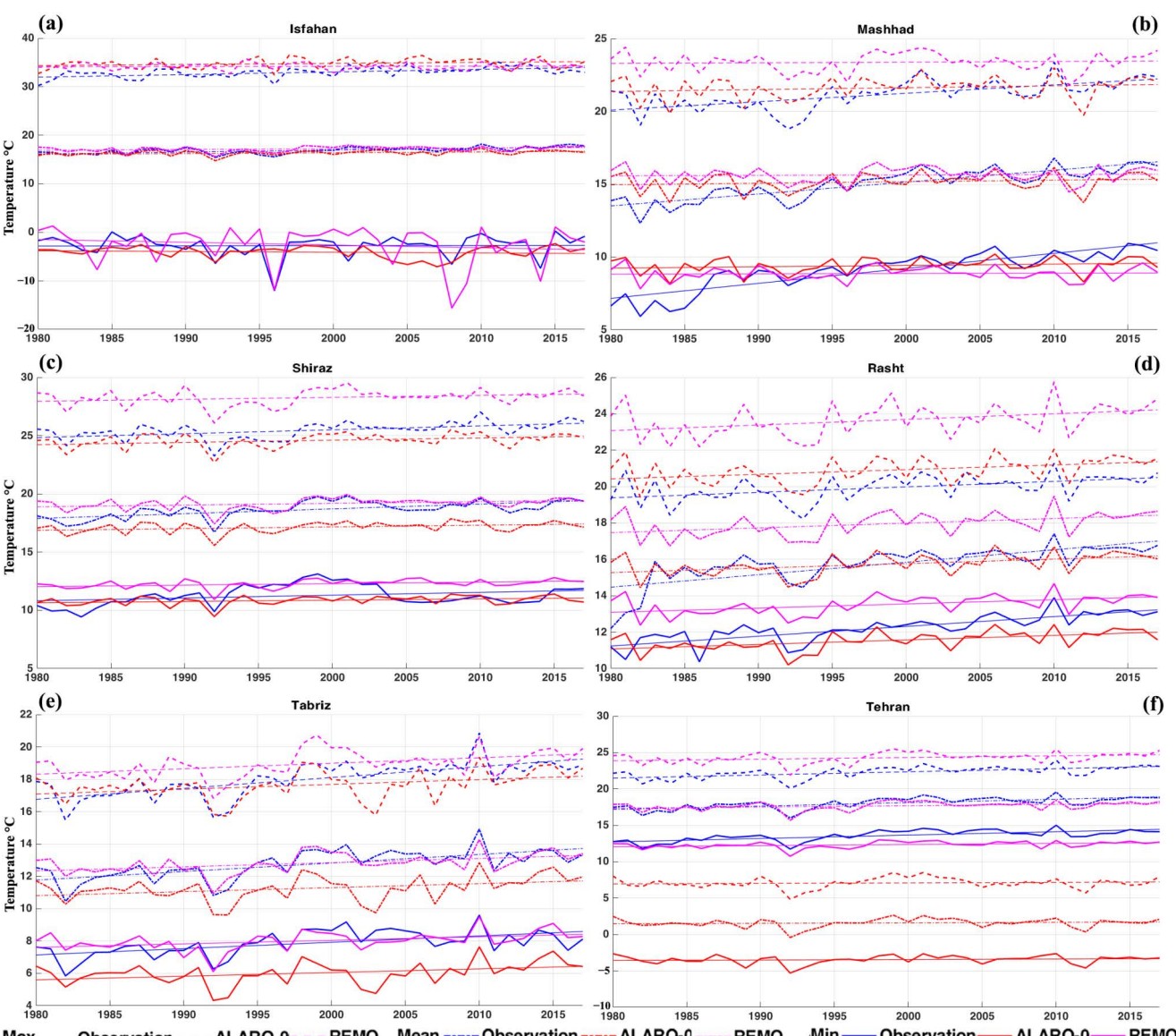

**Figure 2.** Maximum, mean and minimum temperature comparing models with observations for Isfahan (**a**), Mashhad (**b**), Shiraz (**c**), Rasht (**d**), Tabriz (**e**) and Tehran (**f**) during 1980–2017.

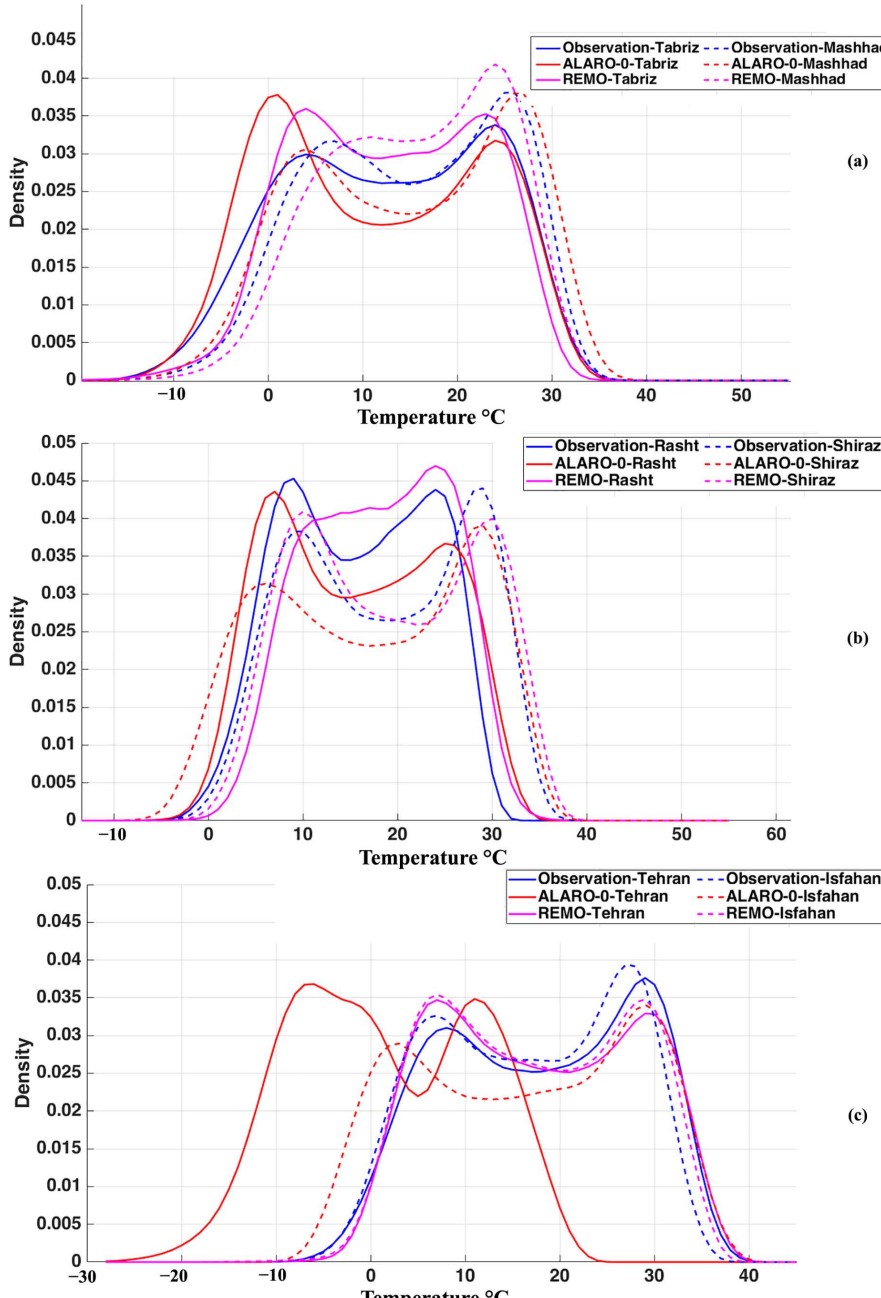

**Figure 3.** Probability density function plot showing distribution density for daily mean temperature comparing models with observations over the studied stations: (**a**) Tabriz and Mashhad, (**b**) Rasht and Shiraz, (**c**) Tehran and Isfahan during 1980–2017.

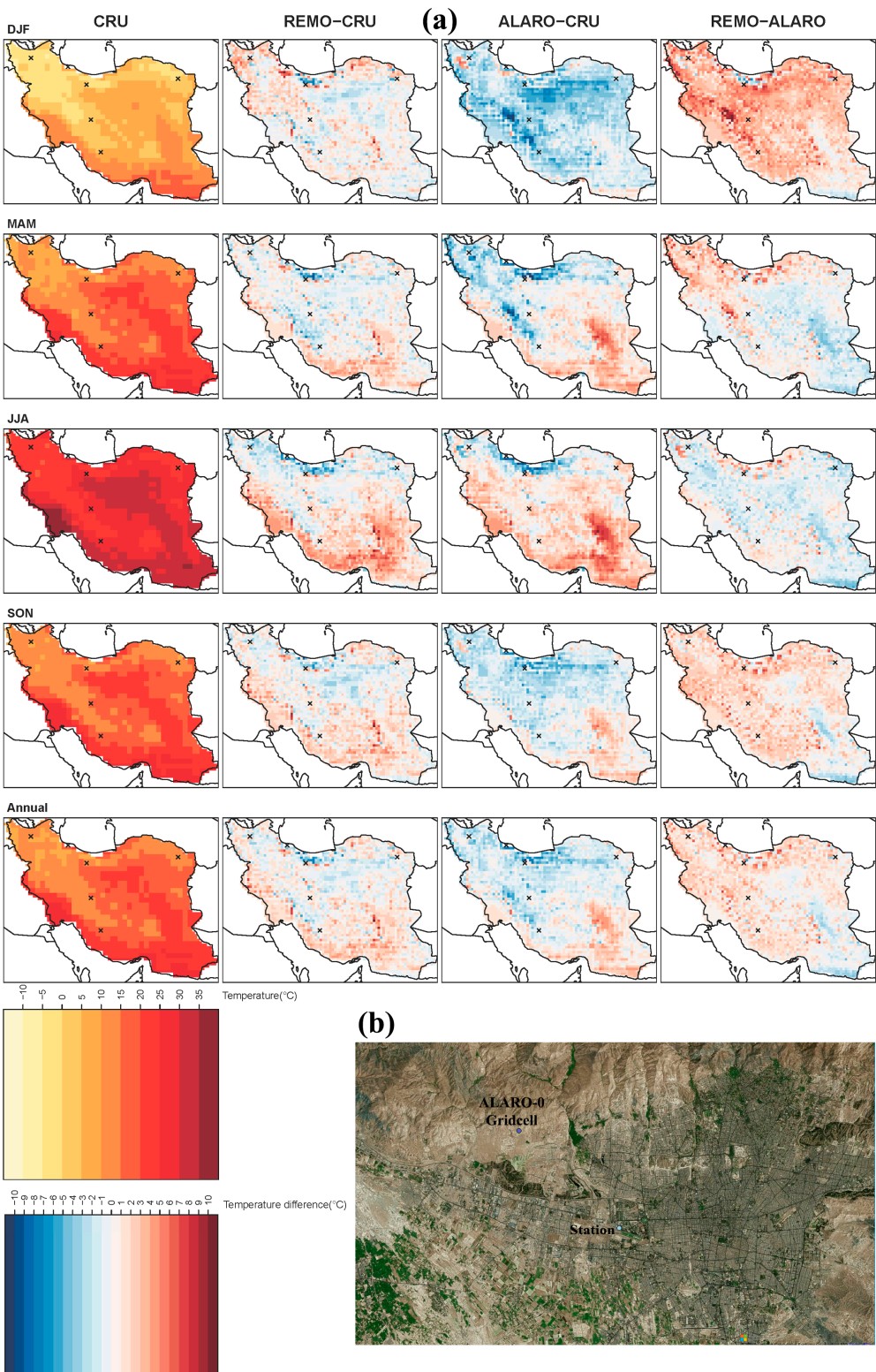

**Figure 4.** (**a**) Spatial representation of temperature over Iran based comparing CRU observational gridded dataset and bias for REMO and ALARO-0 compared to CRU at 0.22°. A very cold bias just above the cross of Tehran is highlighted with dark blue. (**b**) Aerial map of Tehran comparing measurement station (light blue) coordinations with ALARO-0 gridcell (purple).

The annual and seasonal temporal correlations are presented by the Taylor diagrams in Figure 5. REMO simulates maximum temperatures very well at annual resolution for

Isfahan, Shiraz, Tabriz and Tehran since these locations have a high correlation with the observations (>95%), a low RMSD (<1) and a normalised standard deviation close to 1.

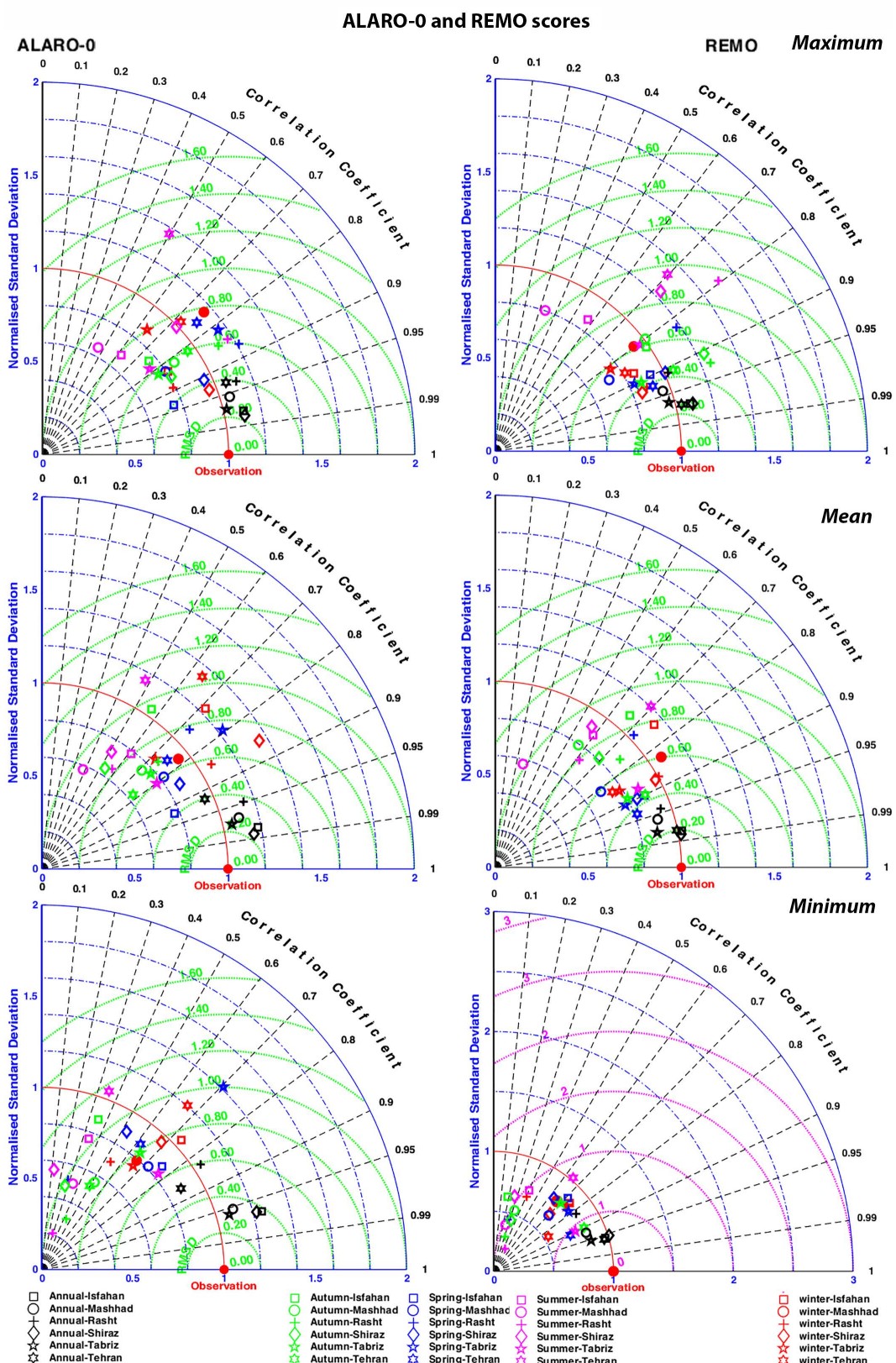

**Figure 5.** Normalised Taylor diagrams for annual and seasonal maximum (**a**), mean (**b**) and minimum (**c**) temperature comparing ALARO-0 and REMO scores for the studied stations during 1980–2017.

ALARO-0 has even better performance than REMO for Isfahan, Shiraz and Tabriz. At the seasonal level, the worst performance is obtained during summer (JJA) for all locations and both models. For mean temperature, the standard deviation is closer to 1, and temporal correlation is higher for REMO than ALARO-0 at annual resolution, except for Tabriz and Mashad. These two locations had the lowest mean temperature over the studied period out of the six locations studied (see Supplementary Materials). Both models also perform worse during the summer (JJA) for mean temperature, except for Tabriz, and for Isfahan in the case of REMO (Figure 3). RMSD is larger for minimum temperature than for mean and maximum temperature over most locations and seasons. Accordingly, the lower performance in minimum temperature negatively influences the performance of mean temperature (Figures 2, 3 and 5). However, at the annual resolution, good performance is still obtained by ALARO-0 for minimum temperature in Isfahan, Shiraz, Tabriz and Mashad. For REMO, minimum temperature values in Shiraz and Isfahan match the observations well at the annual level; the correlation is higher than 95% and RMSD is close to zero.

Figure 6 presents the annual cycles of maximum, mean and minimum temperature for the studied locations. During winter (DJF), ALARO-0 shows a clear cold bias for minimum, mean and maximum temperature and a warm bias for summer (JJA), except for Tehran and Tabriz. For REMO, the temperature deviations in the annual cycle are different for each location.

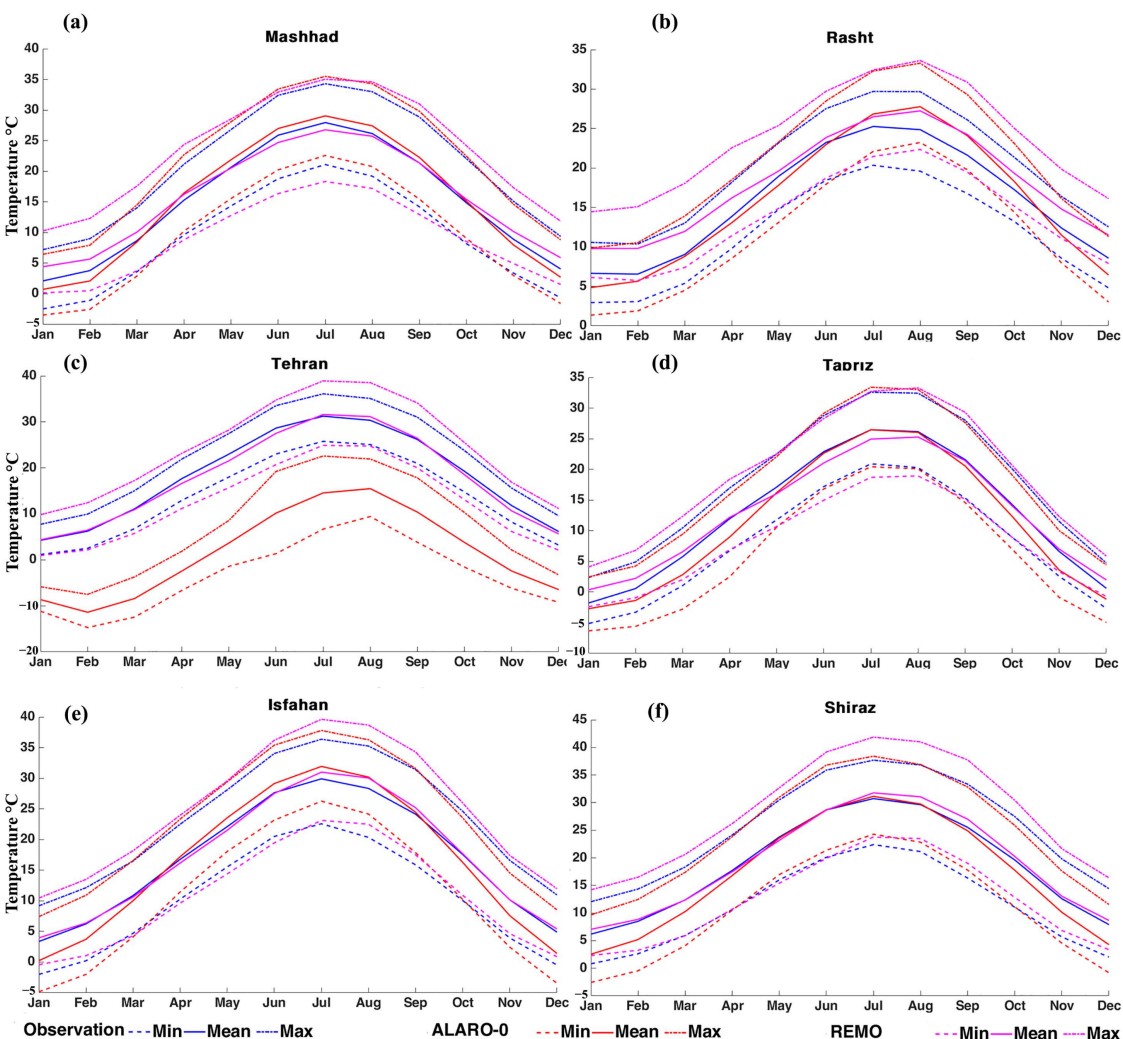

**Figure 6.** Annual cycles for minimum, mean and maximum temperature comparing models with observation for Mashhad (**a**), Rasht (**b**), Tehran (**c**), Tabriz (**d**), Isfahan (**e**) and Shiraz (**f**) during 1980–2017.

### 3.1.2. Freeze–Thaw Cycles

The annual numbers of FTCs during 1980–2017, according to the models and observed data, are presented in Figure 7. Both models are consistent with the observations in following a decreasing trend over time, except for Mashhad and Tabriz. This decreasing trend is due to the general warming trend of annual minimum temperature (Figure 2). Except for Rasht, REMO underestimates the number of FTCs, while ALARO-0 overestimates the number (Figure 7 and Table 3). Moreover, REMO has lower variability in FTCs than the observations, while ALARO-0 has higher variability. The observed FTC values are mainly within the model spread, so when both models are taken into account, the real value is expected to be within this spread.

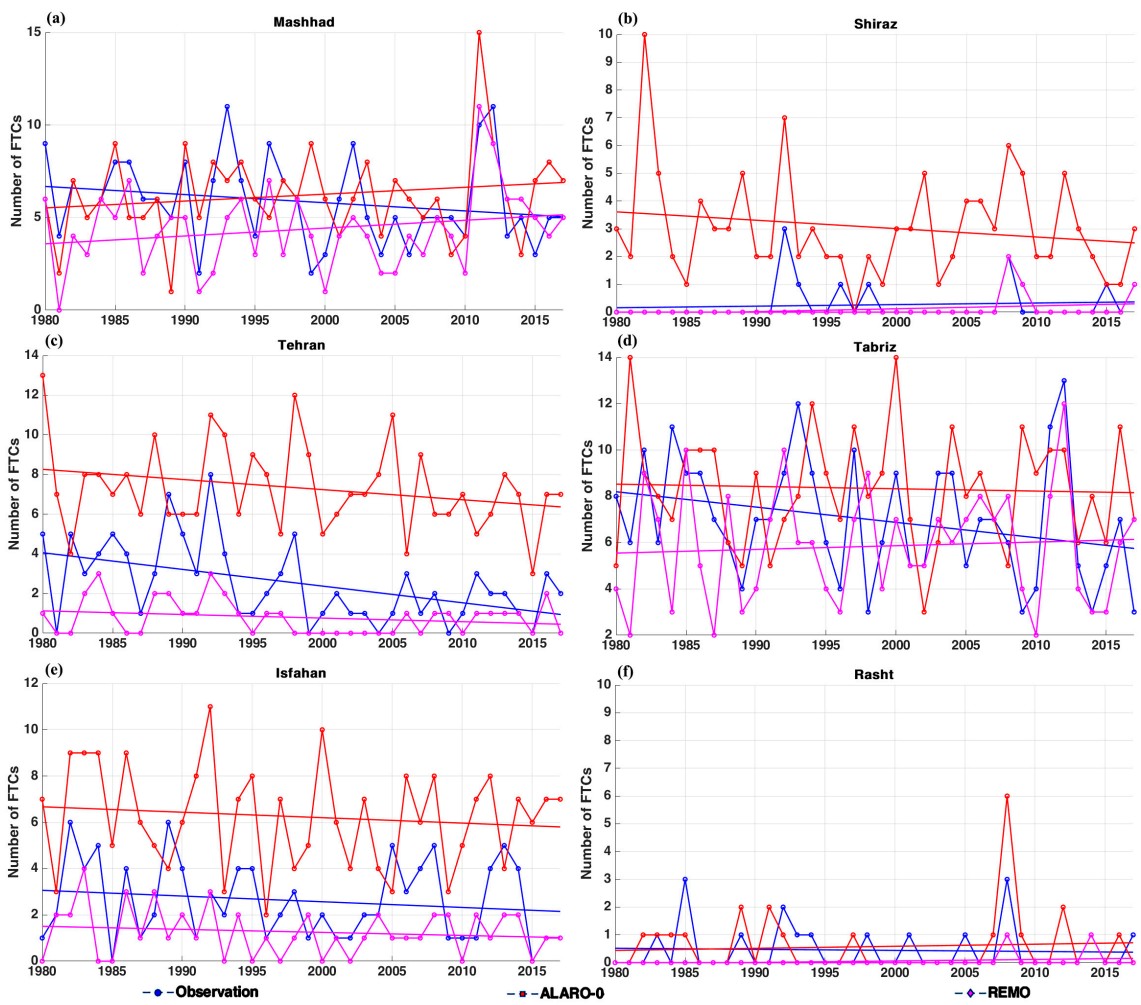

**Figure 7.** The annual number of FTCs for the models and observations over Mashhad (**a**), Shiraz (**b**), Tehran (**c**), Tabriz (**d**), Isfahan (**e**) and Rasht (**f**) during 1980–2017.

**Table 3.** Model performance for FTC index over six studied stations (1980–2017).

|  | Observations | | | ALARO-0 | | | REMO | | |
|---|---|---|---|---|---|---|---|---|---|
|  | Mean of FTCs | Trend Slope | Normalised FTC | Mean of FTCs | Trend Slope | Normalised FTC | Mean of FTCs | Trend Slope | Normalised FTC |
| Isfahan | 2.6 | Decremental | 1 | 6.2 | Decremental | 2.4 | 1.3 | Decremental | 0.50 |
| Mashhad | 5.86 | Decremental | 1 | 6.21 | Incremental | 1.06 | 4.4 | Incremental | 0.75 |
| Rasht | 0.45 | Decremental | 1 | 0.58 | Incremental | 1.3 | 0.5 | Incremental | 1.1 |
| Shiraz | 0.26 | Decremental | 1 | 3 | Incremental | 11.5 | 0.1 | Decremental | 0.40 |
| Tabriz | 7 | Decremental | 1 | 8.3 | Decremental | 1.2 | 5.8 | Incremental | 0.80 |
| Tehran | 2.5 | Decremental | 1 | 7.3 | Decremental | 2.9 | 0.8 | Decremental | 0.32 |

*3.2. Humidity*

3.2.1. Relative Humidity

Figure 8 shows the PDFs for daily average relative humidity over the different locations. The REMO model substantially overestimates low relative humidity values and underestimates high relative humidity values for all sites. A significant underestimation in high relative humidity values can be observed for Rasht, which has the wettest climate among the studied locations (Table 1). ALARO-0 also underestimates the extremely high relative humidity values at this location, but to a lesser extent than REMO. The PDFs of ALARO-0 are more proportional to the observations than those of REMO, except for Tehran, where a substantial overestimation of relative humidity is simulated by ALARO-0. As with temperature, ALARO-0 makes an imperfect estimation of relative humidity for Tehran because the nearest model grid point is located in the mountainous area, outside the urban area, which has a completely different climate. Moreover, there is a notable underestimation of relative humidity in Mashhad for ALARO-0. An overestimation in high relative humidity values can be observed for ALARO-0 at all locations. The normalised Taylor diagrams for relative humidity confirm that both models perform worse for annual average relative humidity in Rasht, since the lowest correlation and highest RMSD are observed for this location (Figure 9). REMO better captures the standard deviation in relative humidity for all places at the annual level, while ALARO-0 has higher correlations and lower RMSD values than REMO, except for Tehran. Comparing the seasons, REMO scores are better during spring (MAM) and winter (DJF). These seasons have typically higher relative humidity for the studied locations. Both models show low performance in predicting relative humidity well during summer (JJA).

The variation in annual relative humidity and the linear trend line are depicted for each location in Figure 10. There is a significant negative bias of the REMO model over the studied stations, whereas ALARO-0 has a limited negative bias for Mashhad, Isfahan and Rasht and a positive bias for Shiraz, Tabriz and Tehran. Except for Rasht, all locations have a decreasing trend in relative humidity observations, and both models correctly predict a decreasing trend. However, for Mashhad and Isfahan, the observed decrease in relative humidity is steeper than the trend of the model outputs. As noted in the previous section regarding temperature, the UHI effect plays a role in these locations. A sharp decrease in relative humidity over time due to the impact of urbanisation has already been extensively reported in the literature [31,32].

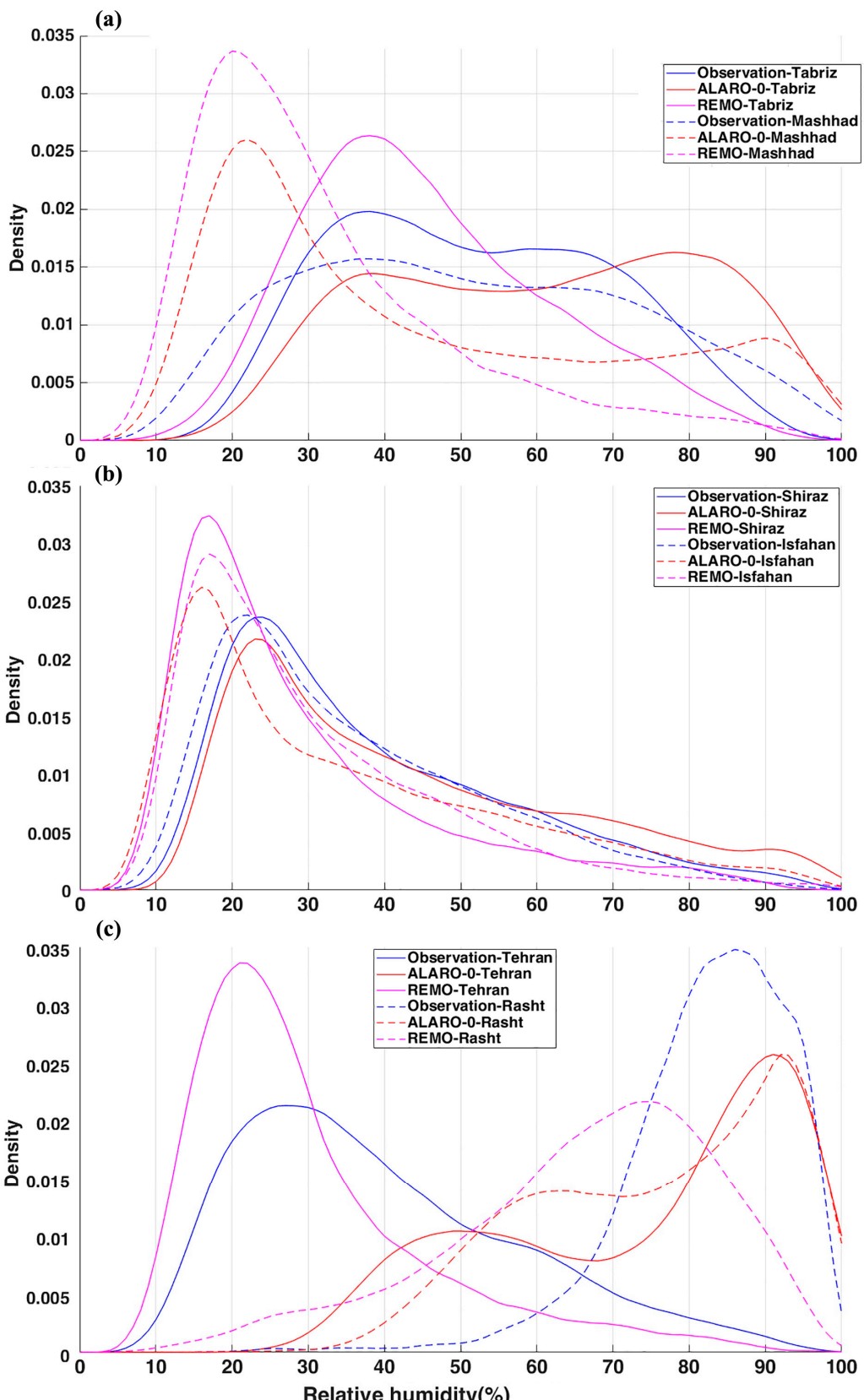

**Figure 8.** PDFs for daily mean relative humidity comparing the ALARO-0 and REMO models with the observations over the studied stations: (**a**) Tabriz and Mashhad, (**b**) Isfahan and Shiraz, (**c**) Tehran and Rasht during 1980–2017.

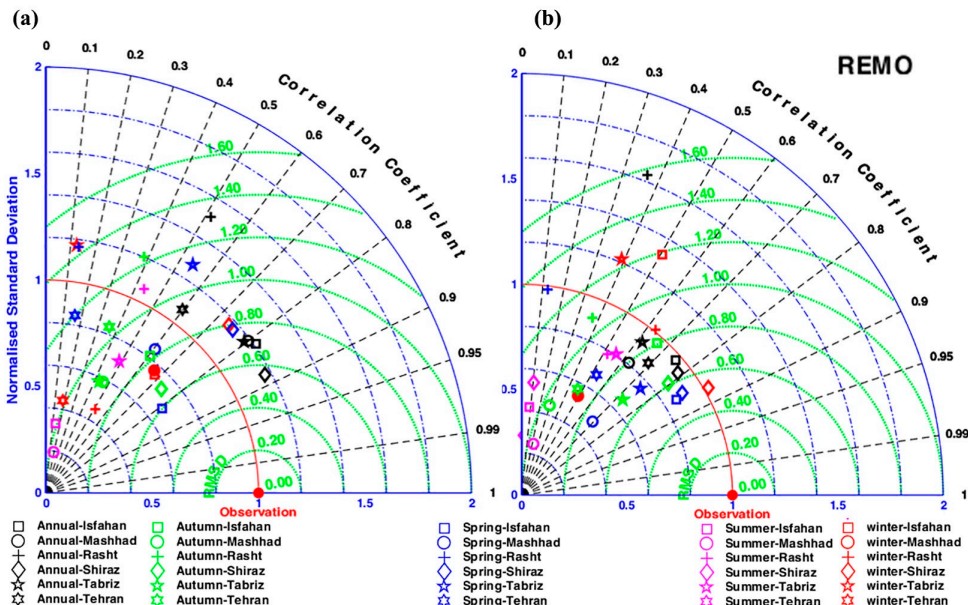

**Figure 9.** Taylor diagrams showing temporal correlation, RMSD and normalised standard deviation comparing ALARO-0 (**a**) and REMO (**b**) for annual and seasonal relative humidity over six studied stations during 1980–2017.

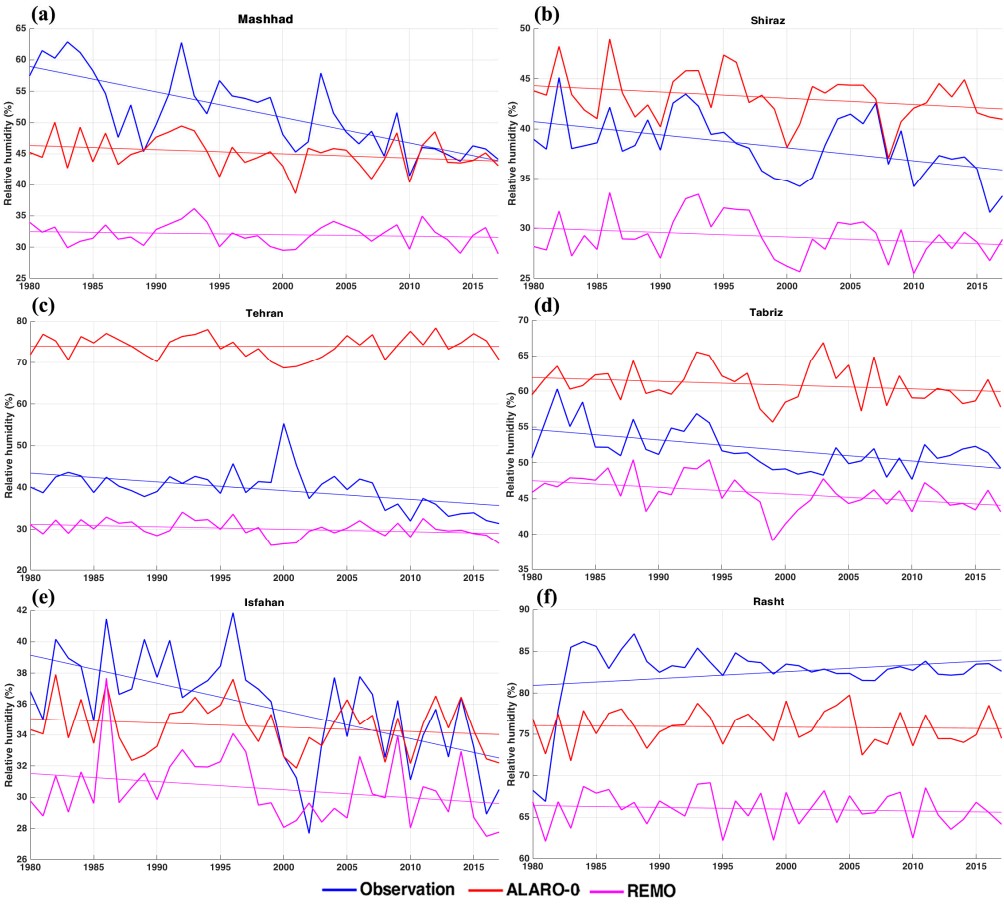

**Figure 10.** Annual average relative humidity for studied stations, comparing models with observations over the studied stations: Mashhad (**a**), Shiraz (**b**), Tehran (**c**), Tabriz (**d**), Isfahan (**e**) and Rasht (**f**) during 1980–2017.

### 3.2.2. Salt Crystallisation Index

Table 4 presents the average observed and modelled annual number of salt transitions and the yearly trend slope for the studied locations. For the number of halite transitions, REMO often predicts within the range of observations, except for Rasht and Mashhad, whereas in these cities ALARO-0 predicts within the spread of the observations. For Rasht, REMO shows a substantial positive bias in the number of transitions for both types of salts. For thenardite–mirabilite transitions, ALARO-0 results always fall within the spread of the observations, and REMO predicts well over Isfahan, Tabriz and Tehran (Table 4).

**Table 4.** Model performance for the annual number of transitions of halite and thenardite–mirabilite (T-M). (N, number of phase transitions; T, trend slope (Year$^{-1}$)). Modelled values marked in green fall within the range of observations (1980–2017).

| | Observation | | | | ALARO-0 | | | | REMO | | | |
| | Halite | | T-M | | Halite | | T-M | | Halite | | T-M | |
| | N | T | N | T | N | T | N | T | N | T | N | T |
| Isfahan | 6.4 | −0.05 | 11.5 | −0.16 | 10 | −0.02 | 14.5 | −0.03 | 6.4 | −0.01 | 10.4 | −0.1 |
| Mashhad | 22.6 | −0.14 | 22.1 | 0.0003 | 21 | −0.03 | 19.7 | 0.004 | 9.3 | −0.03 | 13.7 | 0.03 |
| Rasht | 30.32 | −0.04 | 15.6 | −0.24 | 23 | −0.04 | 14.5 | −0.08 | 51 | −0.07 | 36 | −0.04 |
| Shiraz | 10.4 | −0.12 | 16 | −0.19 | 16 | −0.1 | 11.2 | −0.014 | 14.4 | −0.06 | 8 | −0.05 |
| Tabriz | 18 | −0.12 | 18 | 0.04 | 25 | −0.06 | 15.7 | −0.02 | 13.1 | −0.18 | 23.2 | −0.04 |
| Tehran | 9.6 | −0.16 | 14.2 | −0.2 | 18 | 0.16 | 12.8 | 0.08 | 6.2 | −0.07 | 11.2 | −0.08 |

The equilibrium lines represent the phase boundary of monthly mean relative humidity for halite (75.3% relative humidity) and thenardite–mirabilite transitions in Rasht, at sites with buildings sensitive to salt weathering, are reported in Figure 11a,b. The number of phase changes for this type of salt moves towards zero during warm months. As can be seen, the number of transitions increases when the mean relative humidity gets closer to the equilibrium line point for the phase change. The varying number of transitions over the year indicates different seasonality for the two types of salts (Figure 11b). For the observations, relative humidity decreases during the warm months but stays close to the equilibrium point. As a result, a notable growth trend in halite transitions can be observed from spring (MAM), which stabilises during the summer (JJA) and decreases in early autumn (SON). Relative humidity values produced by ALARO-0 following the same seasonality pattern decrease during warm months, but with a sharper slope that causes a huge difference from the equilibrium line and, as a result, in contrast to the observations, the number of phase changes for halite drops during warm months.

The yearly trend for both kinds of salt transitions during the studied period are presented in Figure 11c–h, showing a decremental trend for the number of halite transitions for both models and observations, except for Tehran, where ALARO-0, unlike the observations, shows an incremental trend for both types of salt. For thenardite–mirabilite transitions, the observed data show an incremental trend over Mashhad and Tabriz, followed by both models over Mashhad, whereas over Tabriz, ALARO-0 shows a decremental trend. A notable negative bias for halite transitions can be observed for REMO over the studied locations, except for Rasht, with a significant warm bias, where both models' annual trend is different from the observations.

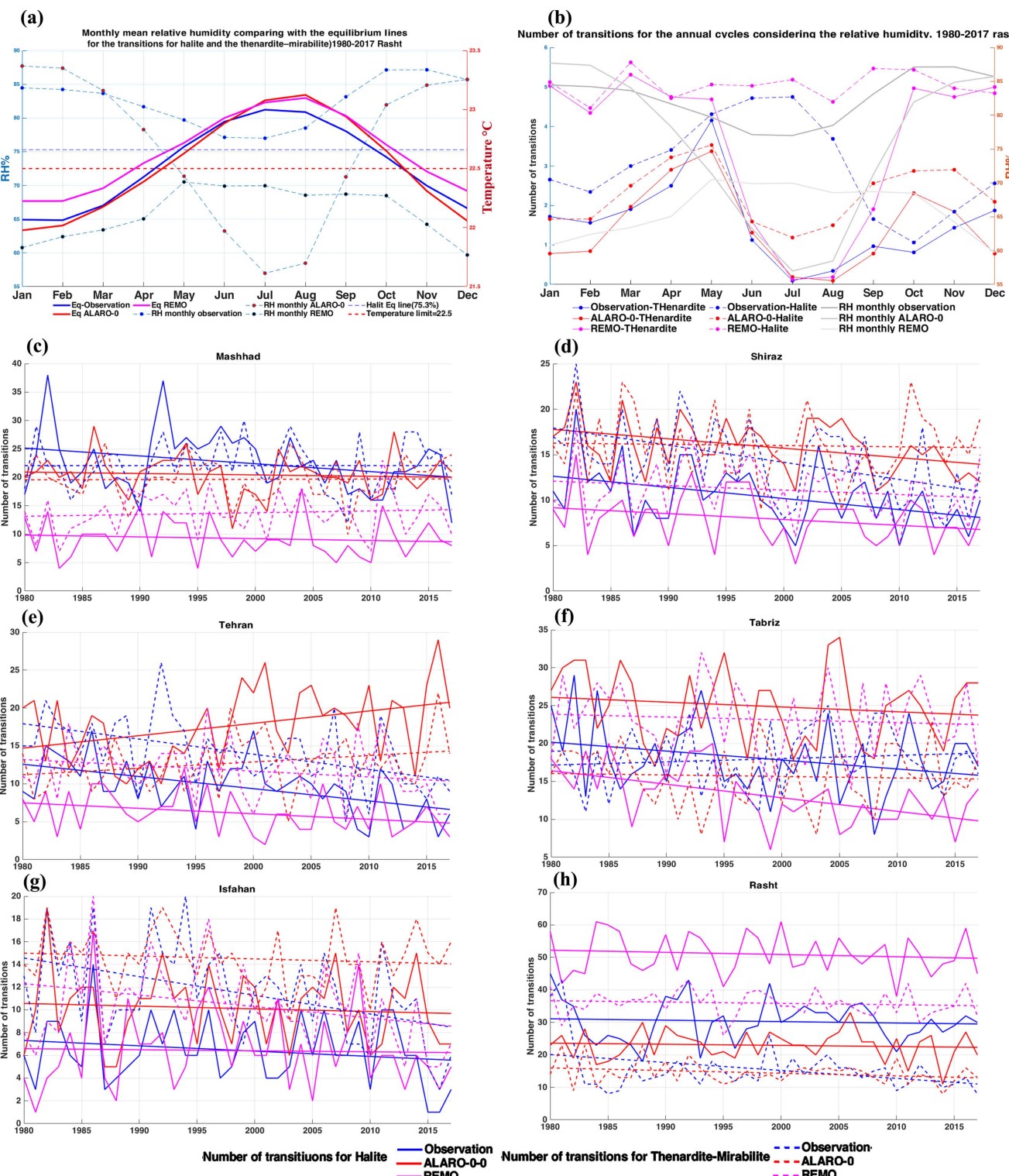

**Figure 11.** (**a**) Mean monthly relative humidity comparing with equilibrium lines of transitions of halite and thenardite–mirabilite for Rasht. (**b**) The number of transitions for annual cycles comparing ALARO-0 and REMO with observations considering relative humidity in Rasht. The yearly number of phase transitions for halite and thenardite–mirabilite comparing ALARO-0 and REMO with observations over the studied stations: Mashhad (**c**), Shiraz (**d**), Tehran (**e**), Tabriz (**f**), Isfahan (**g**) and Rasht (**h**) during 1980–2017.

### 3.3. Wetting and Drying

3.3.1. Precipitation

Figure 12a–c shows the PDF plot for total annual precipitation over the studied area. As can be observed, for Rasht (with the highest yearly precipitation), Shiraz and Mashhad, the REMO distribution is concentrated in the low-range values, thus notably underestimated. A similar consistency with the observations over Isfahan can be observed for the REMO predictions. The ALARO-0 distribution over Mashhad, Shiraz and Rasht is more proportional to the observations than the REMO output while underestimating for Isfahan (with the lowest annual precipitation).

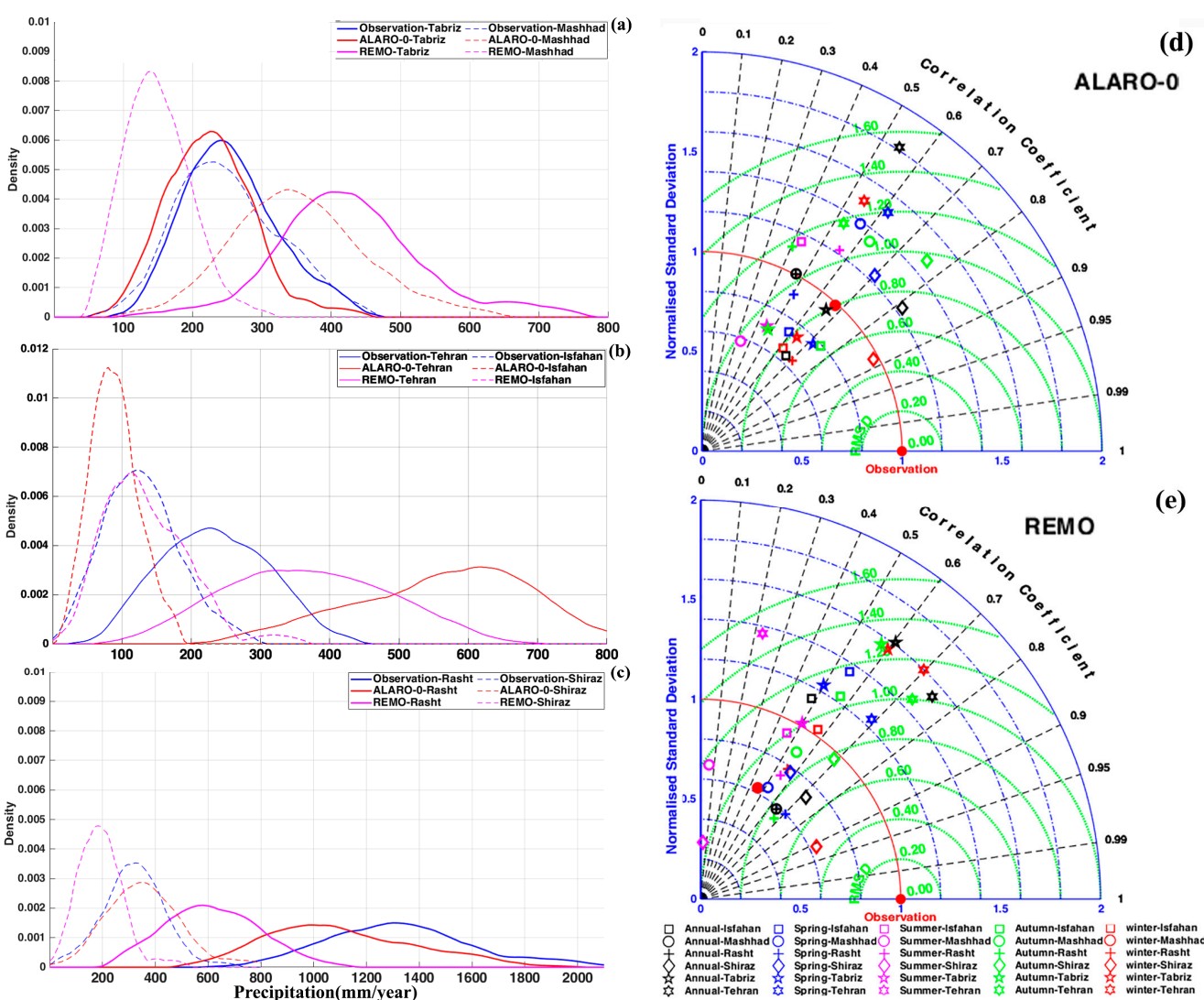

**Figure 12.** PDFs for total annual precipitation comparing ALARO-0 and REMO with observations over the studied stations: (**a**) Tabriz and Mashhad, (**b**) Isfahan and Tehran, (**c**) Shiraz and Rasht during 1980–2017. Taylor diagrams comparing ALARO-0 (**d**) and REMO (**e**) for the annual and seasonal relative humidity over six studied stations during 1980–2017. (**d,e**) Taylor diagrams showing ALARO-0 and REMO scores over six stations for annual and seasonal precipitation.

The PDF analysis results are reflected in the standard deviation of the models from the mean of observations (Figure 12d,e), where the consistency, overestimation or underestimation of the model's distribution can be clearly observed. For the annual resolution, the REMO model shows better performance over Rasht, Mashhad and Tehran. Over Tabriz, Isfahan and Shiraz, ALARO-0 produces data more consistent with the observations. An analysis of the seasonal resolution shows that during the autumn (SON), REMO shows bet-

ter results over Tabriz, Tehran, Rasht and Shiraz, whereas ALARO-0 obtains better scores for Mashhad and Isfahan. For the spring (MAM), ALARO-0 shows better performance over Shiraz, Isfahan and Tabriz. For Mashhad, Rasht and Tehran, REMO output shows better performance during spring (MAM). The winter (DJF) evaluation scores, with the highest precipitation, show that REMO has better performance over Shiraz, Tehran and Mashhad. The ALARO-0 output predicts more precisely over Tabriz, Isfahan, and Rasht (rainiest city). Scores during summer (JJA), with the lowest rainfall, are substantially low. The highest scores are achieved by ALARO-0 during winter (DJF). For REMO, the computed correlations for different regions are concentrated between 0.3 and 0.6. The lowest scores are obtained during summer (JJA).

An analysis of the yearly trends for all 38 years of the study is shown in Figure 10. Except for the trend of REMO over Isfahan and annual ALARO-0 over Rasht, which contrast with the observations, both models follow the observations. For Isfahan, Mashhad, Tabriz and Shiraz, ALARO-0 shows a slight dry bias, whereas over Tehran it has a significant wet bias. The REMO annual trend shows a dry bias over Mashhad, Rasht (significantly drier) and Shiraz, and a wet bias over Tehran and Tabriz (significant wet bias).

### 3.3.2. Moisture Index

The average MI values during the 38 years and the trend's slope for both RCMs are shown in Table 5 based on the annual time series (Figure 13). ALARO-0 output, except for Rasht and Tehran, falls within the range of the observations and follows a decrement similar to observations. REMO output falls within the spread of the observations over Isfahan and Tehran and shows a dry bias for the rest of the locations except Tabriz, where it shows a wet bias. This mainly refers to the model's bias for precipitation. The significant overestimation over Tehran by ALARO-0 refers to overestimation of precipitation and underestimation of temperature, resulting in a notable underestimation of the drying index. Due to minimising the precipitation parameter, both model outputs show a dry bias and inconsistent trend over Rasht as the wettest studied location.

**Table 5.** Model performance for moisture index (MI) (1980–2017). MI mean: average moisture index during 38 years of study (kg air. mm/kg water). Trend slope: slope of trend during 38 years.

|  | Observations | | ALARO-0 | | REMO | |
|---|---|---|---|---|---|---|
|  | MI | | MI | | MI | |
|  | MI Mean | Slope | MI Mean | Slope | MI Mean | Slope |
| Isfahan | 0.34 | −0.001 | 0.22 | −0.001 | 0.34 | −0.006 |
| Mashhad | 0.85 | −0.02 | 1.05 | −0.01 | 0.43 | −0.001 |
| Rasht | 13.75 | −0.15 | 7.3 | 0.0001 | 3.3 | −0.02 |
| Shiraz | 0.8 | −0.01 | 0.94 | −0.02 | 0.42 | −0.005 |
| Tabriz | 1.03 | −0.008 | 1.04 | −0.01 | 1.7 | −0.02 |
| Tehran | 0.6 | −0.008 | 7.35 | 0.001 | 0.86 | −0.007 |

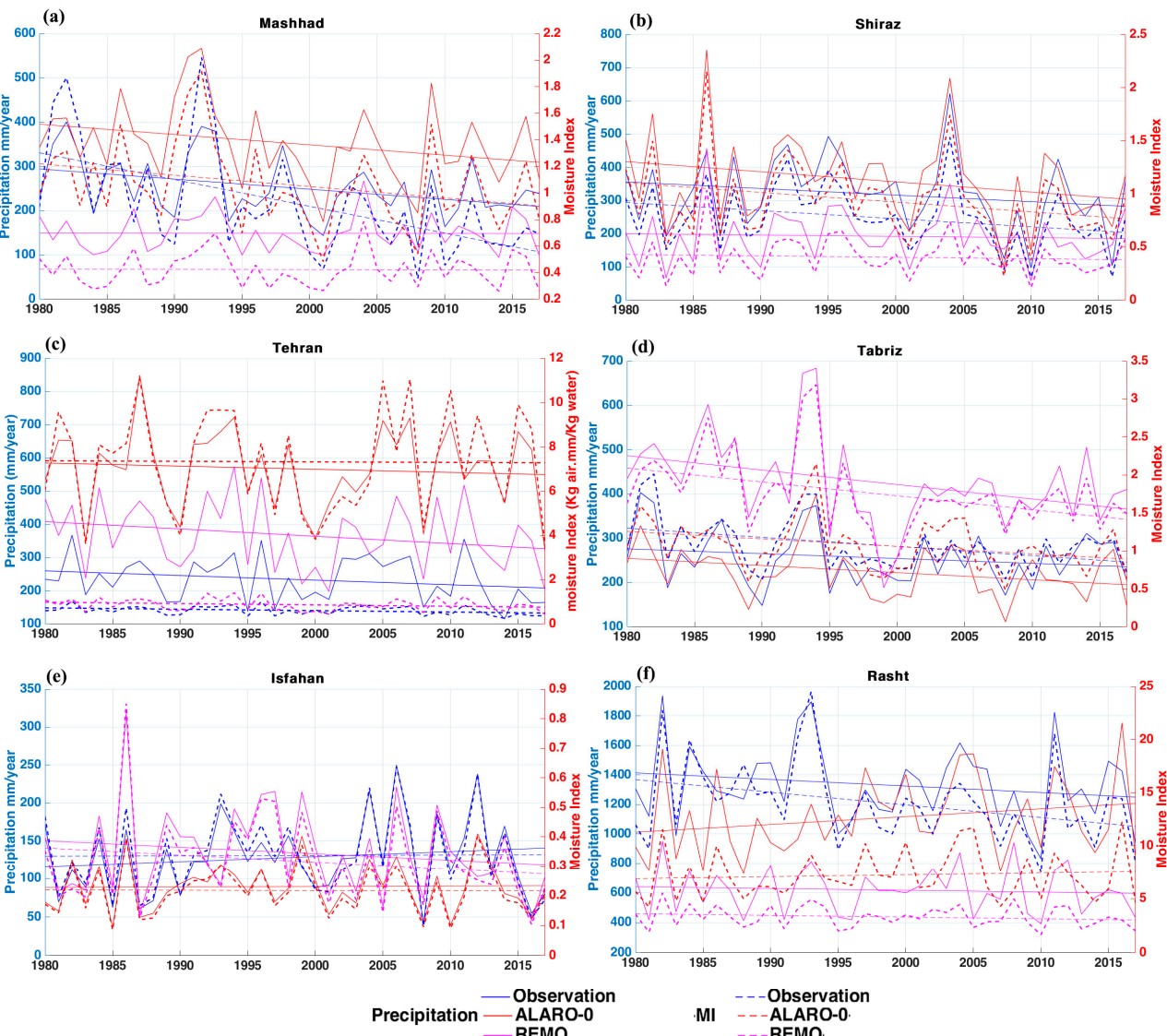

**Figure 13.** Total annual precipitation combined with annual moisture index comparing ALARO-0 and REMO with observations over six studied stations: Mashhad (**a**), Shiraz (**b**), Tehran (**c**), Tabriz (**d**), Isfahan (**e**) and Rasht (**f**) during 1980–2017.

The computed values of the wetting index (WI) and drying index (DI) were used to select wet and dry moisture reference years (MRYs) for each of the 38 years, comparing RCM output with the observations (Figure 14). The plot shows the deviation from the mean values of WI and DI for individual years in terms of standard deviations. Based on this method [27], the years with the largest difference between WI and DI were selected as wet and dry years for models and observations; the year with the largest wet bias represents the wet years and the year with the largest dry bias represents the dry years. The descending trend of annual mean temperature and precipitation for the models and observations clearly shows the drying index's dependency on temperature and relative humidity. The wet year is slightly colder and more humid than the average. Figure 14 shows how wetting and drying affect the reference year selection [27]. The particulars of the MRYs selected and compared with the average climatic values are shown in Supplementary Materials.

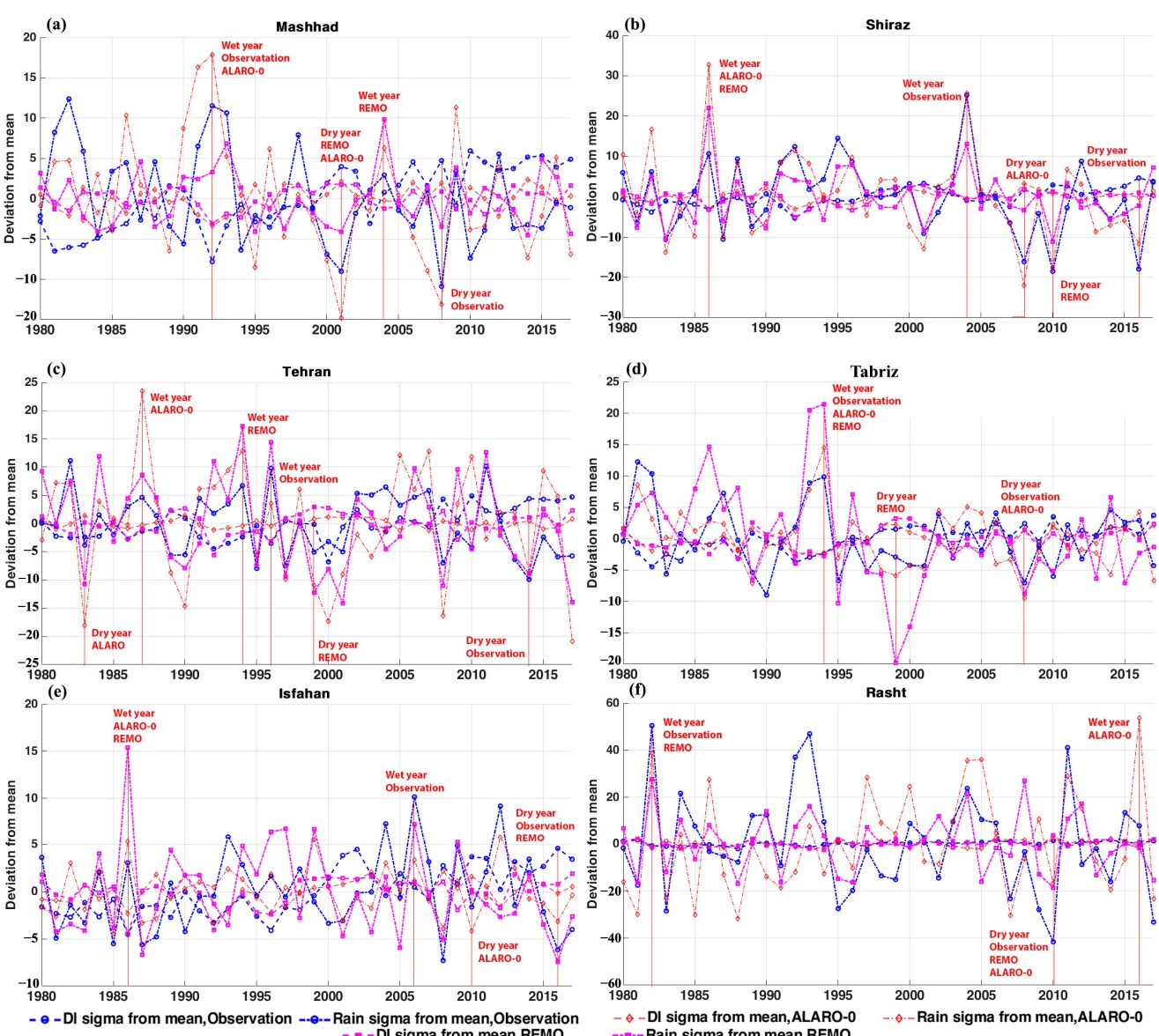

**Figure 14.** Deviation of individual years from means of wet and dry index comparing models with observation over six studied stations: Mashhad (**a**), Shiraz (**b**), Tehran (**c**), Tabriz (**d**), Isfahan (**e**) and Rasht (**f**) during 1980–2017.

*3.4. Wind Parameters*

3.4.1. Wind Velocity and Wind Direction

The PDF plot (Figure 15a–c) shows the distribution density of wind velocity parameter over the studied locations. As can be seen, there is underestimation by ALARO-0 and overestimation by REMO over all captured stations.

Figure 15e,f shows the distribution density plot for wind direction at the stations during the studied period. As can be observed, similar to wind velocity, both models have significant bias. Given that wind direction mainly plays a role in wind-driven rain calculations, this uncertainty affects the accuracy of the building simulations, so in order to increase the accuracy of the simulation, model calibration is recommended. As can be observed for ALARO-0, the distribution of wind direction is spread widely, unlike REMO predictions, which are more concentrated in specific directions. The Taylor diagrams (Figure 16) show the models' performance for average daily and seasonal wind speed. Generally, the correlation and deviation of the model outputs for wind velocity are lower than other climate parameters observed in other model evaluations. ALARO-0's negative

bias is considerably more than REMO's, whereas the correlation of ALARO-0 predictions is slightly higher than REMO's. Accordingly, for Tabriz and Mashhad, the calculated RMSD of ALARO-0 is lower than that of REMO. The annual trend plots show significant overestimation by REMO and underestimation by ALARO-0, and due to considerable differences between simulated PDFs and observations, bias correction of both models for this parameter is recommended.

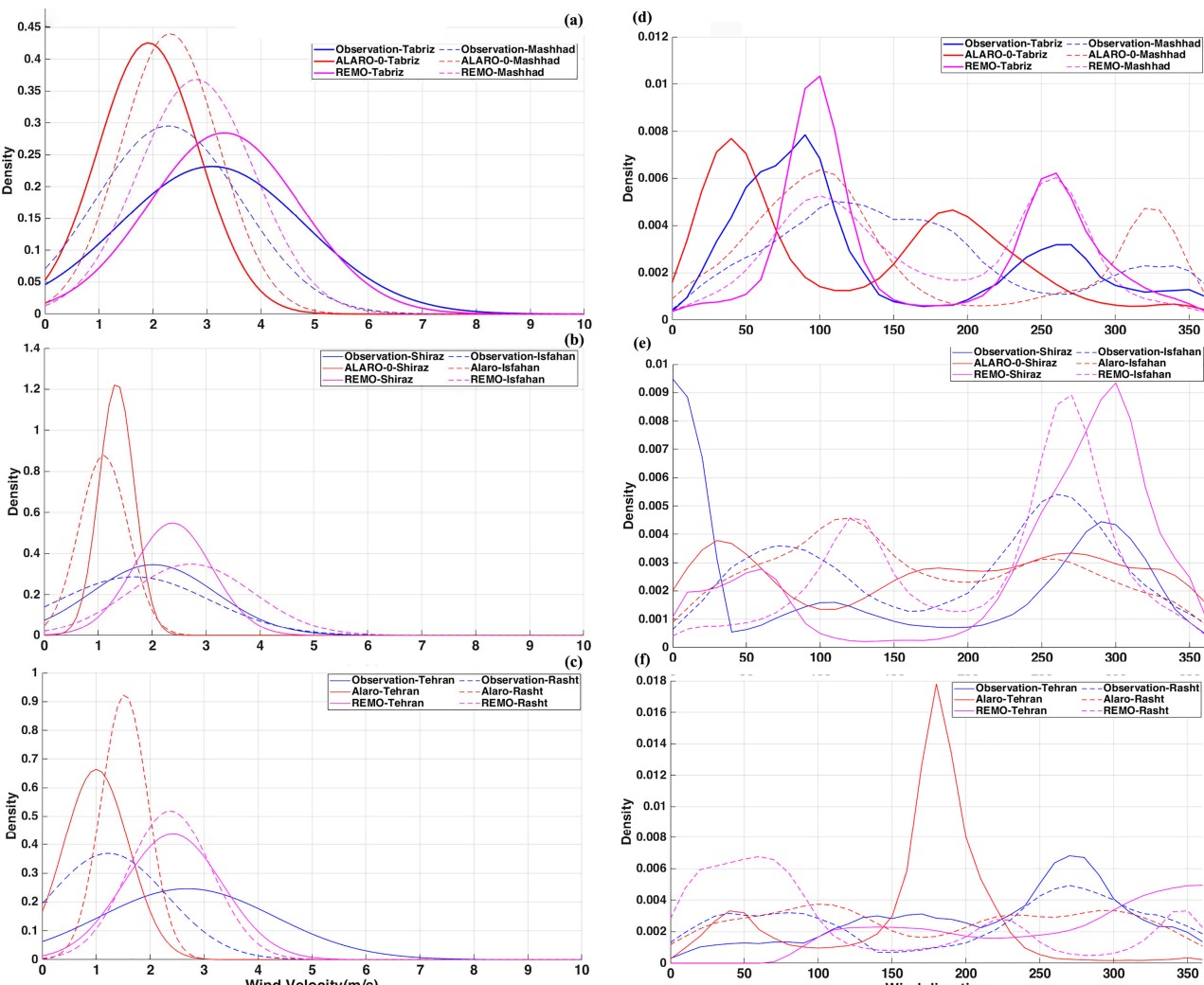

**Figure 15.** (**a**–**c**) PDF plot of average daily wind velocity and (**d**–**f**) six-hourly wind direction comparing ALARO-0 and REMO RCMs with observations over six studied stations: (**a**,**d**) Tabriz and Mashhad, (**b**,**e**) Isfahan and Tehran, (**c**,**f**) Shiraz and Rasht during 1980–2017.

The annual cycle plot (Figure 17) shows significant bias by both models following the PDFs. REMO output for the studied locations always shows a considerable positive bias, whereas ALARO-0 shows a notable negative bias over Tabriz, Shiraz, Isfahan and Tehran. For Mashhad and Rasht, ALARO-0 predictions are overestimated.

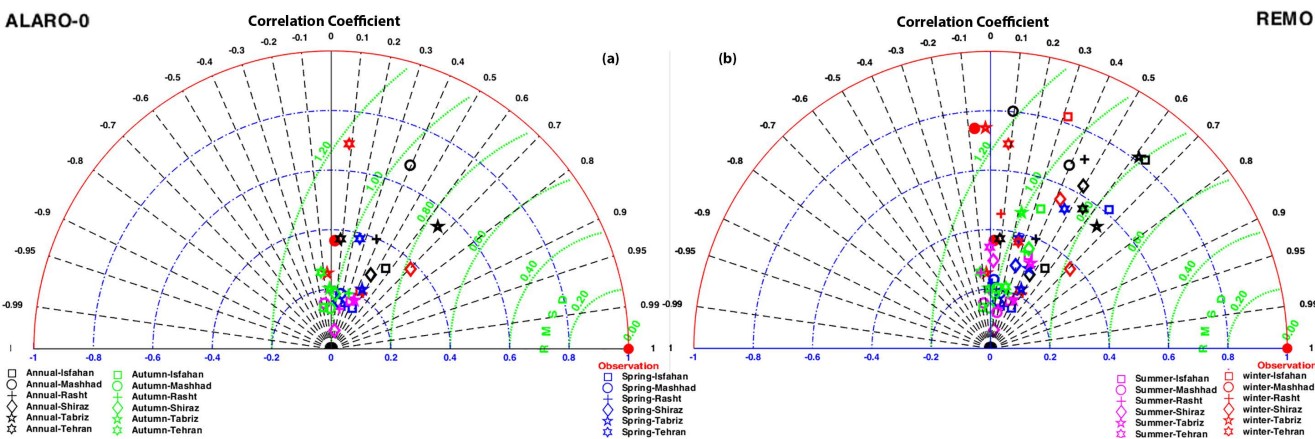

**Figure 16.** Taylor diagram of average annual/seasonal wind velocity comparing ALARO-0 (**a**) and REMO (**b**) models with observations over six studied stations.

**Figure 17.** Annual average wind velocity comparing ALARO-0 and REMO with observations over six studied stations: Mashhad (**a**), Shiraz (**b**), Tehran (**c**), Tabriz (**d**), Isfahan (**e**) and Rasht (**f**) during 1980–2017.

### 3.4.2. Wind-Driven Rain

The total annual wind-driven rain (WDR) computed for the studied sites is reported in Figure 18 and Table 6. As discussed, this parameter is a function of wind speed, wind direction and horizontal rain. The significant bias in both models for wind parameters, particularly wind direction, is reflected in the models' substantial bias for wind-driven rain. For each location, the dominant wind direction was taken into account. As a result of the sharp increase in observed wind velocity, an incremental trend for the computed WDR can be observed for the studied locations except for Shiraz, with a decremental trend. The results of the model outputs differ and are often incompatible with the observations. Over Shiraz, both models show a rather constant WDR load, in contrast to the observed decremental trend. In addition, for Mashhad, Tabriz, Isfahan, Rasht and Tehran, due to the significant bias for wind velocity despite the high accuracy of both models for precipitation, the evaluated models show low performance.

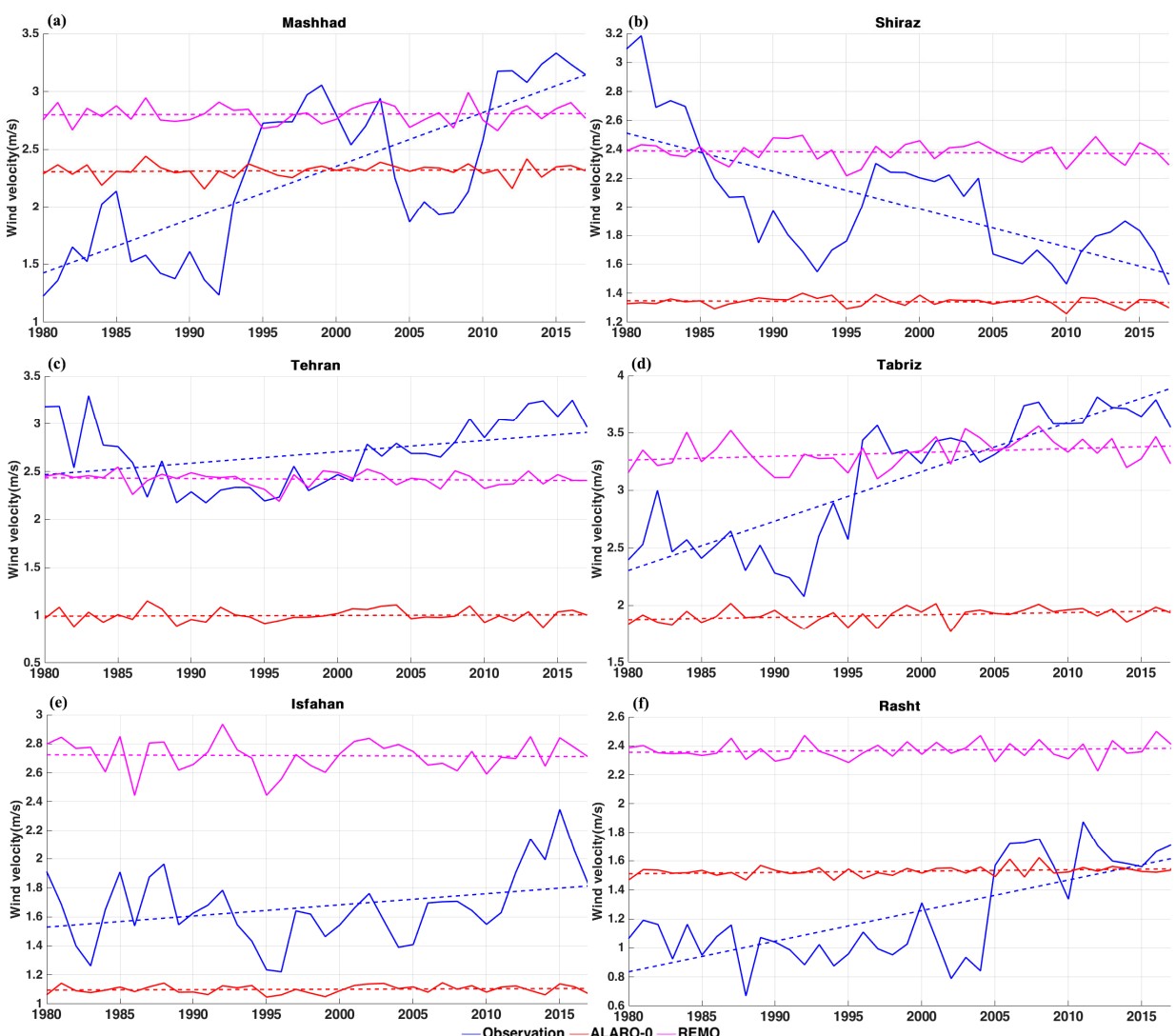

**Figure 18.** Total annual wind-driven rain for dominant wind direction comparing ALARO-0 and REMO with observations over six studied stations: Mashhad (**a**), Shiraz (**b**), Tehran (**c**), Tabriz (**d**), Isfahan (**e**) and Rasht (**f**) during 1980–2017.

**Table 6.** Total annual wind-driven rain comparing models with observations over different locations.WDR: annual WDR (mm/year); slope: trend slope (mm/year).

| | Observation | | ALARO-0 | | REMO | |
|---|---|---|---|---|---|---|
| | Wind-Driven Rain | | Wind-Driven Rain | | Wind-Driven Rain | |
| | WDR | Slope | WDR | Slope | WDR | Slope |
| Isfahan | 18.6 | −0.16 | 7.65 | 0.05 | 28.15 | −0.48 |
| Mashhad | 35.5 | 0.91 | 81.5 | −0.8 | 35.5 | 0.2 |
| Rasht | 285.4 | 0.53 | 88.12 | 0.32 | 37.4 | 0.3 |
| Shiraz | 26.8 | −1.05 | 13 | −0.1 | 50 | −0.8 |
| Tabriz | 51.3 | −1.05 | 35 | −0.1 | 65.5 | 1.3 |
| Tehran | 16.7 | 0.17 | 75.8 | −0.28 | 43.4 | −0.7 |

## 4. Conclusions

In this research, two high-resolution regional climate models, ALARO-0 and REMO, were evaluated over different sites in Iran. Considering the vastness of the Iranian plateau and the diversity of climatic zones, six locations in different climate regions with varying altitude were picked to evaluate RCMs over the whole region. For the first time, in addition to main climatic parameters, some derived parameters and indices critical to buildings, such as FTC, were used for model validation. The evaluation results of the analysed parameters over each studied station are shown in heat maps (Figure 19). Both models reproduce temperature and precipitation features well. As can be observed, the computed RMSD for temperature is generally higher than the other evaluation metrics for both models, showing the models' good performance on temperature. It should be noted that RMSD is clearly higher for minimum temperature than maximum temperature over different locations. Based on the computed RMSD for temperature, the REMO model shows better performance than ALARO-0.

For precipitation, REMO again has better performance over Tehran, Mashhad and Rasht (wettest studied location). The ALARO-0 output outperforms REMO over Shiraz, Isfahan and Tabriz. For relative humidity, except for Tehran, ALARO-0 obtained a higher score.

The analysis of wind velocity shows that ALARO-0 had better performance over Mashhad, Tabriz and Tehran. In the other locations, REMO outperforms ALARO-0.

This study clearly shows that the typical meteorological evaluation based on climatic parameters (comprising air temperature, precipitation and, rarely, wind velocity) should not be considered when selecting a climate model for building simulations. As can be observed, the studied RCMs can predict temperature and precipitation parameters well while showing significant bias for wind velocity and direction.

The resolution of the models, which cannot be adapted to the station's coordinates, affects their reliability. It should be noted that interpolation from 25 km to one single point leads to strong bias, in particular for precipitation. Furthermore, the orography of the station can be important in model evaluation. Applying height correction can reduce the model's bias, but only for the temperature parameter. This difference in model accuracy can seriously affect the model performance on the derived parameters that are essential in building pathology, i.e., FTC and MI. For the annual number of FTCs, ALARO-0 shows better performance over Mashhad, Shiraz and Tehran, whereas REMO produces more consistent numbers over the other locations. For the MI, except for Isfahan (driest location) and Tehran, where ALARO-0 shows a significant negative bias for temperature, ALARO-0 delivers better performance than REMO. Comparing the models for the annual number of salt transitions (halite and thenardite–mirabilite) shows that for Mashhad, Rasht, Tabriz and Shiraz, ALARO-0 predicts better than REMO.

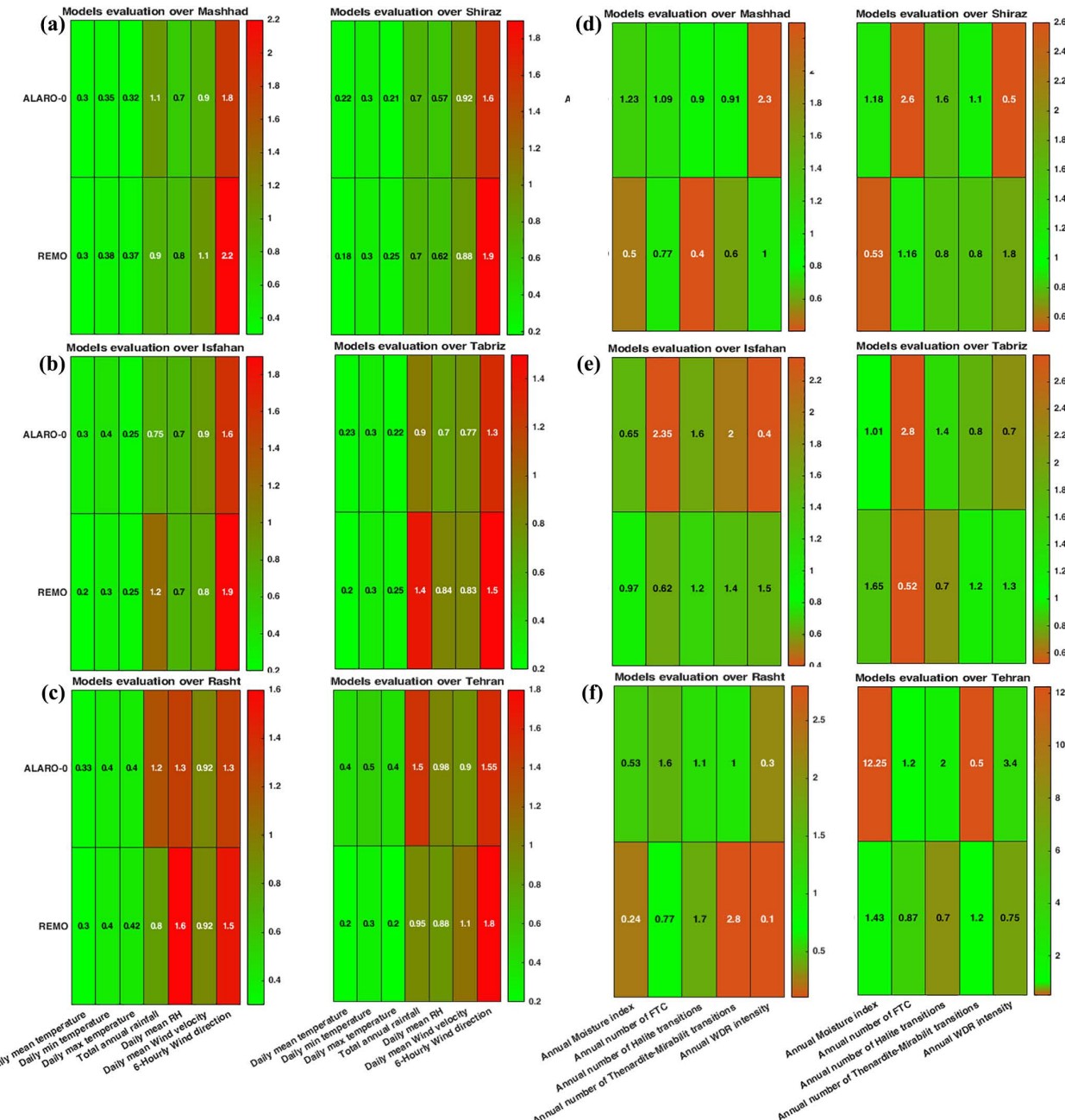

**Figure 19.** (**a–c**) Computed RMSD of models used for evaluating typical climatic parameters and (**d–f**) the normalised average values used for derived parameters over the studied stations: (**a,d**) Mashhad and Shiraz, (**b,e**) Isfahan and Tabriz and (**c,f**) Rasht and Tehran.

Finally, except for Rasht, REMO outperforms ALARO-0 over different locations for total annual wind-driven rain.

Highlighting the importance of time series analysis when evaluating models in research on building physics is another result of this research.

This study indicates that model calibration (bias correction) is required for wind velocity and wind direction parameters over the whole region to get more precise results from building simulations. Finally, it is highly recommended that the proper model be picked based on the evaluation results over the study location. Using coupled ocean-atmosphere models (not only atmosphere, such as REMO), e.g., a combination of ALARO-0 and REMO, might improve the results. Furthermore, the effect of spatial resolution should

be studied to know if a model with a high resolution up to 3–5 km is needed, or lower-resolution models are sufficient for the research scope.

**Supplementary Materials:** The following are available online at https://www.mdpi.com/article/10.339 0/buildings11080376/s1, Table S1: Selected MRYs for the stations, Table S2: Annual average temperature.

**Author Contributions:** Resources, S.C., L.K. and S.T.; N.V.D.B. and M.S.; writing—original draft, H.H.; writing—review and editing, H.H., S.T., S.C., L.K. and N.V.D.B. All authors have read and agreed to the published version of the manuscript.

**Funding:** This research received no external funding.

**Institutional Review Board Statement:** Not applicable.

**Informed Consent Statement:** Not applicable.

**Data Availability Statement:** The climate data produced by ALARO-0 and REMO2015 were uploaded to the ESGF data nodes (website: http://esgf.llnl.gov/, last access: 7 July 2020). In order to obtain the data, choose a node, click on or search for "CORDEX", then select the domain "CAS-22" and the RCM in the left column.

**Acknowledgments:** The authors would like to acknowledge the Iran Meteorological Organisation for providing the climate data for the studied stations. The computational resources and services for the ALARO-0 regional climate simulations were provided by the Flemish Supercomputer Center (VSC) and the EWI department of the Flemish government. The CORDEX-CORE REMO simulations were performed under the GERICS/HZG share at the German Climate Computing Centre (DKRZ).

**Conflicts of Interest:** The authors declare no conflict of interest.

## Nomenclature

| | |
|---|---|
| CRU | Climatic research unit |
| FTC | Freeze–thaw cycle |
| MI | Moisture index |
| GCM | General circulation model |
| RCM | Regional climate model |
| RCP | Representative concentration pathway |
| DJF | December, January, February |
| MAM | March, April, May |
| SON | September, October, November |
| JJA | June, July, August |
| RMSD | Root mean square deviation |
| PDF | Probability density function |
| MRY | Moisture reference years |
| HAM | Heat–air–moisture |
| UHI | Urban heat island |
| WDR | Wind-driven rain |

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
