# Peer review of "Evaluation of ALARO-0 and REMO Regional Climate Models over Iran Focusing on Building Material Degradation Criteria"

_buildings, doi:10.3390/buildings11080376_

Round 1

Reviewer 1 Report

The article compares observational data and two regional climate models for the territory of Iran. Detailed description of air temperature, precipitation, wind speed and direction, as well as indices for assessing the influence of climate on buildings was done in the paper.

Below are the comments on the article:

Line 78: the text of the article does not indicate why these two RCMs were chosen

Line 121: Please add study period to the table title

Figure 4. On the figure 4 the CRU data were used, but in the Materials and Method section this data source not mentioned. 

All data used in the work should be clearly written the section Materials and Method.

Line 320: this is figure 7, not 6. Lines on the figure are poorly visible. 

Figure 9: the correlation coefficients are truncated in the diagrams 

Figure 10: the figure title is written in italics

Line 399: this is table 4

Figure 11 is not visible

There are a lot of figures in the article. Consider optimizing your drawings, maybe some of them are unnecessary

Reviewer 2 Report

General comments:

The paper describes the evaluation of two regional climate models (ALARO-0 and REMO) focusing on effects of climatic conditions on building materials.

I don't understand the meaning of this evaluation.

Do you want to check if the considered models could reproduce the effect of the future climate on building materials?

Do you want to check the models performances in comparison with observed values?

I suggest to better explain what it is yours purpose.

Specific comments:

lines 248-251: where are the nearest grid points for these locations?

section 3.2.2: The section should be better explained, can you better explain the meaning of 'equilibrium line' (line 376)?

line 399: I think table 4 instead of table 3

Figure n. 11 is not visible in the text

lines 616 - 617: the sentence is difficult to understand

line 622: 'coupled models', what kind of model can be considered?

In the supplementary file sheet named 'MRYs': the years of observations, ALARO and REMO are different (for example: 2016, 2010, 2016; or 2016, 2008, 2010). I think it is difficult to understand and connect to the text. 

In the supplementary file sheet named 'Sheet1':  It is difficult to understand and connect to the text the comparison of different years.

Reviewer 3 Report

The review on the manuscript in journal Buildings entitled „Evaluation of Regional Climate Models ALARO-0 and REMO over Iran focusing on building material degradation criteria“.

The article analyzes the Regional Climate Models of ALARO-0 and REMO in Iranian conditions to assess building material degradation criteria.

Research methods have been described at a more or less satisfactory level. Statistical analysis of the data needs some improvement and addition.

The main problem is that the writing of the article is eclectic and the technical implementation does not comply with the basic principles of academic writing.

The conclusions are based on analysis and are more or less adequate.

Broad comments

The calculation methods should be described in detail in the "Materials and Methods" section, not in "Appendix A".

Line 162 reads "the amplitude of their variations that is represented by their standard deviation from the mean of the observations". The standard deviation has a correct meaning only if the parameter under analysis has a normal distribution. It would be necessary to specify whether and by what method (test) the compliance with the normal distribution of parameters has been checked. The correspondence of the distribution of data for a specific period to the normal distribution should be checked, for example, by Shapiro - Wilk, Jarque - Bera, Smirnov - Kolmogorov and other tests.

In the figures showing the linear trend lines (Figures 2, 6, 10, 13, 14) it would be correct to add the corresponding confidence bounds.

Figures and Tables must be numbered in the order in which they appear. All Tables and Figures must have references in the article in the text that precedes the Table or Figure. However, the article has two Table 2 (line 153 and line 399). Table 4 and Table 6 are missing from the article, although the reference to Table 4 is in the article (line 368) and the article is Table 7 (line 566). The article has two Figure 6 (line 301 and line 320). There is no Figure 7 in the article, but the reference to Figure 7 is in the article (line 305).

The reference list must contain only those reference entries to which the article refers. The article does not refer to sources 31 and 32.

It is not necessary (not correct) to repeat the figure caption text in the graphic area of the figure. This note applies to all figures except Figure 4.

In the case of Figures consisting of several parts, it is necessary to mark all the parts (for example (a), (b), etc.). Explanations of all parts must be in Figure caption only. When referring to different parts of the figures, their relative position (right, left (lines 254, 518, ), top, middle (line 255), bottom) is not used, but the symbols of the parts (for example (a), (b), etc.) are used.

Academic writing should be objective. If it is subjective or emotional, it will lose persuasiveness and may be regarded as relying on emotion rather than building a reasonable argument based on evidence. The language or informal writing should therefore be impersonal, and should not include personal pronouns. For most subject areas the writing is expected to be objective. For this the first person (I, we, me, my, etc.) should be avoided. In this article on lines 72 and 73 are written „we need“, on line 75 is written „we will“, etc. lines 209, 210, 526, 605, 623, 624, 661, 649, 666 ,681. Eliminating personal pronouns from writing is highly recommend.

The article needs technical revision.

Specific comments

The caption of the Figure must be below the Figure - but line 233 is above the Figure.

Figure 11 is not shown correctly. (A) and (B) are partially visible, part (C) is completely absent.

Figure captions must be presented in the same style - line 366 is unlike other formatted italics.

Equations (lines 653, 656, 662, 671, 676, 679, 684) must be fully created using the Equation Editor. The equations must be centered and the equations numbers aligned to the right.

It must always be a space between the numerical value and unit symbol except the plane angle and percent – line 154 „10m“, line 378 „22.5°C“, etc.

When referring to several sources at a time, only one bracket should be used - line 52 "[5] [6] [7] [8] [9] [10] [11] [12] [13]" should be [5-13], etc.

When referring to Tables and Figures, the same style should be used in the article – line 240 „Figure 4“, line 250 „Fig. 3“, line 258 „figure 5“, etc. When referring to a specific Table or Figure in an article, the reference must begin with a capital letter - line 478 "table 5", etc.

There are missing spaces between words and other parts of the sentence in many places - line 118 "environments [2]", line 126 "(2013) [18]", line 136 "locations (Table 2)", line 197 "method [27] ", line 436" Figure11", etc., etc.

Round 2

Reviewer 3 Report

The review on the reviewed manuscript in journal Buildings entitled „Evaluation of Regional Climate Models ALARO-0 and REMO over Iran focusing on building material degradation criteria“.

The article analyzes the Regional Climate Models of ALARO-0 and REMO in Iranian conditions to assess building material degradation criteria.

Research methods have been described at a more or less satisfactory level. Statistical analysis of the data needs some improvement and addition.

The main problem is that the writing of the article is eclectic and the technical implementation does not comply with the basic principles of academic writing.

The conclusions are based on analysis and are more or less adequate.

Comments

The graphic quality of all the drawings (Figure 1 to Figure 19) is very poor.

Line 28 – Reference source not found.

The references in the article must be all in one style. Line 197 is a reference to the author in the author referencing style, the reference itself is not in the Reference list.

The reference list contains two references 16, lines 804 - 806 having the same text.

Figure 1 caption is repeated in the text on lines 137, 138.

Figure 4 caption is repeated in the text on lines 370-372.

Equations (lines 253, 256, 282, 288, 292, 303) must be fully created using the Equation Editor. The equations must be centered and the equations numbers aligned to the right.

After line 510, an incomprehensible combination of an unnumbered table and a figure in Figure 14.

There are lot of missing spaces between words and other parts of the sentence in many places - line 69 " structures[14]", line 90 „Iran[16]", line 111 "significance[17]", line 155 " (2013)[18]", line 156 (2016)[19]", etc., etc.

When referring to sections, the same style should be used in the article – line 114 „Section 3.1“, line 116 „section 3.3“, line 124 „section 3.3“, etc. When referring to a specific section in an article, the reference must begin with a capital letter.

Line 338 „Figure 2. shows“ - point redundant.

Line 471 - end point of sentence inside a sentence.

Lines 556 and 575 have references to Figures, but Figures numbers are missing.

Lines 560 have "Kg", which is not the correct abbreviation for kilogram.

In line 256, the temperature unit is incorrect.

This article requires a major technical adjustment.

Round 3

Reviewer 3 Report

The review on the reviewed manuscript in journal Buildings entitled „Evaluation of Regional Climate Models ALARO-0 and REMO over Iran focusing on building material degradation criteria“.

The article analyzes the Regional Climate Models of ALARO-0 and REMO in Iranian conditions to assess building material degradation criteria.

Research methods have been described at a more or less satisfactory level. Statistical analysis of the data needs some improvement and addition.

The main problem is that the writing of the article is eclectic and the technical implementation does not comply with the basic principles of academic writing.

The conclusions are based on analysis and are more or less adequate.

Comments

Line 24 – Reference error.

Line 93 - A period and a space at the beginning of a sentence.

Line 115 „Table 1.It“ - There is no space at the end of the sentence.

Line 130 – „climate(1980-2017)“ - There is no space after the word "climate".

The Table title and Figure caption must be in the same style in the article, this also applies to the endpoint: In Figures 1, 7, 9 and 16 there is no dot at the end of the caption, in Figures 2-6, 8, 10-15, 17-19 there is a dot; For Table 1, 4, 6 there is a dot at the end of the title, there is no dot at the end of the Table 2, 3, 5 title.

The references in the article must be all in one style. Line 135 is a reference to the author in the author referencing style. "(2013)" is redundant, there is no space between words. The same reference errors are on line 136.

Line 197 „cycles(FTC)“ - there is no space between the words.

Line 199 - temperature unit incorrect - is „0° C“, but correct is „0 °C“.

Line 209 – „damage[25]“ - there is no space between the words.

Line 209 – „chloride(Halite)“ - there is no space between the words.

Line 217 – „MPa[26]“ - there is no space between the words.

Line 223 – „MPa[27]“ - there is no space between the words.

Line 228 – „????and, ??ℎ?are“ - there are no spaces between the words.

Line 229 – „thenardite(Benavente et al., 2008)“ - incorrect referencing style, there is no space there either.

Line 233 – „[27]so“ - there is no space between the words.

Line 234 – „index(MI)“ - there is no space between the words.

Line 247 – „index(WI)“ - there is no space between the words.

Line 252 – „index(WI)“ - there is no space between the words.

When referring to a specific section in an article, the reference must begin with a capital letter. – line 253 „section 3“.

Line 269 – „rain(WDR)“ - there is no space between the words.

Line 305 – „bias(Figure“ - there is no space between the words.

Line 306 – „reason:. ALARO-0“ - one punctuation redundant.

Line 310 - „dataset(Figure“ - there is no space between the words.

Figure 4 places the sentences in the middle of lines 311, 312, which do not fit.

Line 321 – „summer(JJA)“ - there is no space between the words.

Line 325 – „here(see“ - there is no space between the words.

Line 368 – „stations(1980-2017)“ - there is no space between the words.

Caption of Figure 8 „loca-tions“ - redundant punctuation.

Line 380 - a forced line break inside a sentence without any need.

Line 404 – „[32-32]“ - a very dubious and completely pointless reference.

Line 439 – „observations(1980-2017)“ - there is no space between the words.

Capture of Figure 11 – „Rasht.1980-2017“ - suspicious point, no space; „stations(c-f)“ – there is no space between the words.

Line 451 – „han(with“ - there is no space between the words.

Capture of Figure 12 – „locations(a-c)“ and „precipitation(d-e)“ - there is no space between the words.

Line 488 – „observation(Figure“ - there is no space between the words.

Title of Table 5 – „index(MI)(1980-2017)“ and „study(kg“ – there are no spaces between words.

Line 522 - a forced line break inside a sentence without any need.

The figures must be numbered in the order in which they are presented, but the article is preceded by Figure 16 to Figure 15.

Line 537 – „rain(WDR)“ - there is no space between the words.

Line 541 - a forced line break inside a sentence without any need.

Line 550 – „slope(mm/year)“ - there is no space between the words.

The text of the article must refer to all Figures, there is no reference to Figure 19 in the text.

The text of the article must refer to all sources in the Reference list, the text does not refer to sources 16, 22 and 31.

The text of the article must refer to all Tables, there is no reference to Table 3 in the text.

This article requires a thorough technical adaptation. Order corrections from MDPI specialists.
